# Codon optimization underpins generalist parasitism in fungi

Thomas Badet[1†], Remi Peyraud[1†], Malick Mbengue[1†], Olivier Navaud[1], Mark Derbyshire[2], Richard P Oliver[2], Adelin Barbacci[1], Sylvain Raffaele[1*]

[1]LIPM, Université de Toulouse, INRA, CNRS, Castanet-Tolosan, France; [2]Centre for Crop and Disease Management, Department of Environment and Agriculture, Curtin University, Perth, Australia

**Abstract** The range of hosts that parasites can infect is a key determinant of the emergence and spread of disease. Yet, the impact of host range variation on the evolution of parasite genomes remains unknown. Here, we show that codon optimization underlies genome adaptation in broad host range parasites. We found that the longer proteins encoded by broad host range fungi likely increase natural selection on codon optimization in these species. Accordingly, codon optimization correlates with host range across the fungal kingdom. At the species level, biased patterns of synonymous substitutions underpin increased codon optimization in a generalist but not a specialist fungal pathogen. Virulence genes were consistently enriched in highly codon-optimized genes of generalist but not specialist species. We conclude that codon optimization is related to the capacity of parasites to colonize multiple hosts. Our results link genome evolution and translational regulation to the long-term persistence of generalist parasitism.

**\*For correspondence:** sylvain. raffaele@toulouse.inra.fr

[†]These authors contributed equally to this work

**Competing interests:** The authors declare that no competing interests exist.

## Introduction

The host range of a parasite has a central influence on the emergence and spread of disease (**Woolhouse and Gowtage-Sequeria, 2005**). There is a clear demarcation between specialist parasites that can only infect one or a few closely related host species and generalists that can infect more than a hundred unrelated host species (**Woolhouse et al., 2001**; **Barrett et al., 2009**). The host range is a trait constrained by ecological and physiological factors and determined by the evolutionary history of a parasite and its potential hosts (**Poulin and Keeney, 2008**). Host specialization is a frequent pattern in living systems that was proposed to result from tradeoffs in performance on multiple hosts (**Joshi and Thompson, 1995**). At the molecular level, optimization by natural selection of parasite proteins functioning in the interaction with a given host may be detrimental to their function in another species (**Dodds and Rathjen, 2010**; **Dong et al., 2014**). Nevertheless, generalists able to thrive on a broad range of hosts are found in most parasite lineages. The molecular mechanisms associated with the evolution of generalist parasitism remain, however, elusive. Notably, the extent to which interaction with multiple hosts affects the evolution of parasite genomes remains unknown.

Genomic analyses of obligate specialist parasites revealed massive losses of enzymes unnecessary for growth and reproduction on living hosts (**Schirawski et al., 2010**; **Spanu et al., 2010**). Instead, such specialists rely largely on a sets of small secreted protein (SSPs) subverting host cell function. For instance, the genome of the powdery mildew pathogen *Blumeria graminis* f. sp. *hordei* harbors 180 SSPs of about 100–150 amino acids (**Pedersen et al., 2012**). SSPs of specialist fungi often show evidence of diversifying-selection leading to extensive changes in their protein sequence (**Stukenbrock et al., 2011**; **Pedersen et al., 2012**; **Hacquard et al., 2013**). By contrast, the secretion of a battery of degrading enzymes is crucial for the parasitic success of generalists (**Hu et al.,**

2014; Kubicek et al., 2014). The genomes of generalist fungi notably contain in average ~3 times more carbohydrate activity enzymes (CAZYmes) than specialist fungi (Zhao et al., 2013). For instance, Rhizoctonia solani AG2-2IIIB contains over 1000 CAZYme genes (Wibberg et al., 2016). Protein translation is the largest consumer of energy during cellular proliferation, and because secreted proteins are not recycled like other cellular proteins, strong constraints exist to reduce the synthetic cost of secreted proteins in microbes (Smith and Chapman, 2010). Considering the distinct properties of secreted proteins in specialist and generalist parasites, the impact of protein translation efficiency on the cost of secreted protein synthesis likely differ in these organisms.

The differential use of synonymous codons in a genome, or codon usage bias, affects gene expression level, protein folding, translation efficiency and accuracy (Plotkin and Kudla, 2011). In particular, the co-evolution of codons with the genomic tRNA complement, leading to codon optimization, results from the combination of neutral processes and selection for the optimization of protein translation (Hershberg and Petrov, 2008; Shah and Gilchrist, 2011). Natural selection for the optimization of protein translation, through ribosome pausing time, translation error rates and co-translational protein folding, has been widely associated with codon usage biases (Drummond and Wilke, 2008; Tuller et al., 2010; Shah and Gilchrist, 2011). Because the probability of protein translation error increases with protein length, selection for translation accuracy can be higher in genes encoding long proteins, as was observed in E. coli (Eyre-Walker, 1996). Codon usage bias also increases with the evolutionary age of genes, the frequency of codons optimal for translation being significantly higher at codons for conserved amino acids (Marais and Duret, 2001; Prat et al., 2009). Considering that highly expressed secreted proteins are often longer and more conserved in generalist than in specialist parasites, we expect natural selection on the optimization of codons to be stronger in generalist parasites than in host specialists.

The cost of protein translation leads to a tradeoff between protein production and cell growth rate (Scott et al., 2010; Shah et al., 2013; Kafri et al., 2016). Codon optimization therefore impacts cell growth in selective environments (Botzman and Margalit, 2011; Krisko et al., 2014), and could positively impact on the performances of parasites regardless of the host they encounter. This hypothesis would be consistent with generalist lineages performing better on average on their set of available hosts than co-occurring specialist competitors to persist in the biota, as predicted by the 'jack of all trades, master of none' theoretical model (Futuyma and Moreno, 1988). Here, we have examined the strength of natural selection on the optimization of codons in parasites with contrasted host range across the fungal kingdom. Because the genomes of broad host range parasites generally encode longer proteins, an in silico model of the cell translation machinery predicts that natural selection on codon optimization should be stronger in these species. Consistent with this prediction, we found that codon optimization at the genome scale correlates with fungal parasites host range. High codon optimization and broad host ranges co-evolved multiple times independently through fungal phylogeny. To document the molecular bases of codon optimization at the species level in generalist fungi, we compared single-nucleotide polymorphism (SNP) patterns in natural populations of a generalist and a specialist fungal parasite. We detected signatures of purifying natural selection acting on optimal codons in the generalist parasite only. Finally, we show that genes associated with virulence are better codon-optimized in generalist than in specialist parasite genomes supporting codon optimization as an adaptation to the colonization of multiple hosts. Together, our results reveal patterns of adaptive genome evolution associated with generalism that are conserved through the fungal Kingdom. Furthermore, we establish a link between translational regulation and host range variation through genomic patterns of codon optimization, contributing to our understanding of the dynamics of disease epidemics.

## Results

### Long proteins encoded by the genome of generalist fungi likely increase natural selection on codon optimization

The efficiency and accuracy of protein translation constrains cell growth rate and depends on sequence properties of expressed proteins (Ingolia et al., 2009; Kafri et al., 2016). To evaluate the impact of translation efficiency on cell growth during host colonization, we calculated theoretical maximal growth rates for cells of generalist and specialist fungal parasites. For this, we designed an

in silico model of the cell translation machinery describing the accumulation over time of ribosomal proteins, other intracellular proteins and secreted proteins from which the maximal cell growth rate can be deduced (see 'Materials and methods', *Figure 1—source data 1*). To calculate the typical protein synthesis rate, the model uses the median length of ribosomal, intracellular and secreted proteins as input. We determined these values in the complete predicted proteomes of 13 host-specialist fungi (less than four host genera) and 15 generalist fungi (over 10 host genera) all belonging to different genera and covering most major fungal clades (*Figure 1A*, *Figure 1—source data 1*). In average, secreted proteins were ~14.8% longer in generalists than in specialists (365 and 318 codons respectively; Welch's t-test p<$10^{-08}$). Intracellular proteins were also longer in average in generalists than in specialists (~10.1% longer with 381 and 346 codons respectively; Welch's t-test p<$10^{-08}$), whereas ribosomal proteins were slightly shorter in average in generalists than in specialists (~1.2% shorter with 189 and 192 codons respectively; Welch's t-test p<$10^{-04}$). In yeast, codon decoding rate varies from 10 to 28 codons per second, with an average of 20 codons per second (*Gardin et al., 2014*). We tested the impact of varying the average codon decoding rate from 10 to 28 codons per second on the maximal cell growth rate predicted by our model for generalist and specialist fungal cells (*Figure 1B*). Cells of specialists achieved higher maximal growth rates than cells of generalists at a given codon decoding rate. Reciprocally, cells of generalists require higher codon decoding rate to reach maximal growth rates similar to cells of specialists. For instance, in order to achieve a growth rate of 4000 cells per 24 hr, the codon decoding rate should be 21.7 codons per second in generalists but only 19.9 codons per second in specialists.

To test the robustness of these observations across the fungal phylogeny, we compared related specialist and generalist species from three major fungal groups: Basidiomycetes, Ascomycetes and Basal Fungi (*Figure 1C*). In all three groups, secreted proteins were longer in average in generalist species than in specialist species. For instance, secreted proteins of the generalist *Botrytis cinerea* were in average ~73% longer (394 amino acids) than secreted proteins of the specialist *Blumeria graminis* f. sp. *tritici* (228 amino acids; Welch's t-test p<$10^{-08}$). Length differences in ribosomal and other intracellular proteins were not consistent through all three clades. Overall, our cell translation machinery model showed in all three cases that generalist species require higher codon decoding rates to achieve similar growth rates as their specialist relatives.

The in silico cell model used here suggests that longer proteins, especially secreted proteins, encoded in the genome of generalist fungi limits maximal cell growth rates compared to specialist fungi. It also shows that codon optimization, leading to higher codon decoding rates, can support the secretion of more complex proteins with limited growth penalty. Although additional factors may contribute, we show that longer protein pools are sufficient to increase the constraints on codon optimization in generalist fungi. We therefore expect natural selection on the optimization of codons to be stronger in generalist parasites than in host specialists.

## Codon optimization correlates with fungal parasites host range

To test this hypothesis, we analyzed patterns of codon optimization in the genomes of 36 fungal parasite species, including the 15 generalist parasites and 13 specialists analyzed previously, as well as 8 species with intermediate host range (between 4 and 10 host genera) (*Table 1*). For this, we first used a model that infers the degree 'S' of coadaptation between codon usage and the genomic tRNA complement at the whole genome scale (*dos Reis et al., 2004*). S values ranged from −0.032 (*Puccinia graminis*) to 0.843 (*Cryptococcus neoformans*), with an average of 0.374. We performed a correlation analysis and found that codon optimization at the whole genome scale increased with host range (Spearman rho = 0.82, p=7.1 $10^{-10}$, *Figure 2A*). Generalist species had an average S of 0.58, whereas the average S was 0.23 for specialists. We verified the significance of the correlation between host range and genome scale codon optimization using Felsenstein's phylogenetic independent contrasts, and obtained Spearman rho of 0.6 (p=1.5 $10^{-05}$).

Comparing genes codon usage with codon usage for a reference set of highly expressed genes is a classical approach for estimating codon usage bias independently of knowledge of the tRNA repertoire. We determined the codon adaptation index (CAI) (*Sharp and Li, 1987*) using ribosomal protein genes as a reference set. To allow comparisons of codon optimization across species, we calculated for each species the degree '$S_{CAI}$' of correlation between codon usage and CAI at the whole genome scale. $S_{CAI}$ ranged from −0.187 (*Blumeria graminis* f. sp. *tritici*) to 0.87 (*Cryptococcus neoformans*), with an average of 0.438 (*Figure 2B*). $S_{CAI}$ increased with host range (Spearman

**Figure 1.** Contrasted length distribution in proteomes is expected to increase selection on codon optimization in generalist fungi. (**A**) Distribution of length (number of codons) in the ribosomal, intracellular and predicted secreted proteomes of 13 specialist fungal parasites and 15 generalist fungal parasites. (**B**) Relationship between codon decoding rate (number of codons translated per second) and cell growth rate of typical specialist and generalist fungi predicted by a cellular model of protein translation. Dotted lines highlight the higher codon decoding rate required in generalist fungi compared to specialist fungi to achieve a growth rate of 4000 cells produced per day. (**C**) Distribution of length in the proteomes (left) and relationship between codon decoding rate and cell growth rate (right) in related specialist and generalist fungi from the Basidiomycetes, Ascomycetes and Basal Fungi. The width of boxplots is proportional to the number of values. Spe., specialist (green); Gen., generalist (brown). Welch's t-test p: *$<10^{-01}$, **$<10^{-04}$, ***$<10^{-08}$.

*Figure 1 continued on next page*

*Figure 1 continued*

The following source data is available for figure 1:

**Source data 1.** Equations forming the mathematical model of protein biosynthesis related to protein length and codon optimization parameters; list of parameters and variables used for modeling of growth rate based on proteome properties; and values of parameters used for modeling of growth rate based on proteome properties.

rho = 0.61, p=9.0 $10^{-06}$ under phylogenetic independent contrasts) and generalist species had an average $S_{CAI}$ of 0.61, whereas the average $S_{CAI}$ was 0.26 for specialists. O'Neill *et al.* recently developed the self-consistent normalized Relative Codon Adaptation index (scnRCA) as a measure of codon usage bias that does not rely on a reference gene set and corrects for mutational biases (*O'Neill et al., 2013*). We calculated for each species the degree '$S_{scnRCA}$' of correlation between codon usage and scnRCA at the whole genome scale. $S_{scnRCA}$ ranged from −0.085 (*Nosema ceranae*) to 0.853 (*Cryptococcus neoformans*), with an average of 0.495 (*Figure 2C*). $S_{scnRCA}$ increased with host range (Spearman rho = 0.59, p=2.7 $10^{-05}$ under phylogenetic independent contrasts) and generalist species had an average $S_{scnRCA}$ of 0.67, whereas the average $S_{scnRCA}$ was 0.32 for specialists.

To control for the confounding effects of heterogeneity in the completeness of genome assemblies, we first verified that codon optimization was related to the number of hosts but not genome assembly parameters at ANOVA p<0.01 (*Figure 2—source data 1*). Comparing codon usage bias for gene sets conserved across species is another established method to compare codon optimization across species independent of genome completeness. We identified a set of 1620 core ortholog genes (COGs) conserved across all the 36 fungal parasite species (*Figure 2—source data 2*). The tRNA adaptation indices (tAIs) were significantly higher in the orthologs of generalist genomes (median value is 0.361) than in orthologs of specialist genomes (median value is 0.337; Welch's t-test p<0.01) (*Figure 2D*). We calculated S for COGs in each genome and found that S is significantly higher for COGs in generalist genomes (median is 0.774) than in specialist genomes (median is 0.167; Welch's t-test p=9.9 $10^{-06}$) (*Figure 2D*). These different analyses converge toward the conclusion that codon optimization at the whole genome scale correlates with fungal parasites host range.

## Codon optimization and host range co-evolved multiple times across fungal phylogeny

To document the evolution of codon optimization in fungi, we reconstructed the ancestral state of codon optimization in fungal phylogeny. To this end, we built a phylogeny of our 36 fungal parasites and 9 non-parasitic fungi distributed among all major clades covered in our study (*Figure 3A*, *Figure 3—source data 1* and *2*). We determined the degree of codon optimization in non-parasitic fungal genomes and found that codon optimization was generally lower in non-parasitic fungi than in their generalist relatives (average values for non parasitic fungi were S = 0.157, $S_{CAI}$ = 0.315 and $S_{scnRCA}$ = 0.354). This does not exclude that some lineages of non-parasitic fungi could have evolved high codon optimization, but supports the view that codon optimization increased after the divergence of generalist parasites. To support this conclusion, we inferred the ancestral degree of codon optimization at internal nodes of fungal phylogeny using ML ancestral state estimation (*Revell, 2012*) (*Figure 3A*). The fungal parasites species used in this work represented four independent evolutionary paths to extreme generalism (over 100 host genera) with non-parasitic or specialists in the same clade, in the Chytridiomycota, the Agaricomycotina, the Leotiomycetes and the Sordariomycetes (*Figure 3A*). In all four clades, generalism was associated with an increase in genomic codon adaptation compared to related non-parasitic or specialist species and to the reconstructed ancestral state. For instance, in the Leotiomycetes, the generalist parasites *Botrytis cinerea* and *Sclerotinia sclerotiorum* (>300 host genera) both had S > 0.5, whereas the non-parasitic *Oidiodendron maius* and the host specialists *Erysiphe necator* and *Blumeria graminis* f. sp. *tritici* had S < 0.2. The reconstructed ancestral state for this clade was S = 0.37.

To test for the robustness of the association between host range and codon optimization across fungal phylogeny, we tested whether it could be detected at the sub-Kingdom level. The correlation coefficient between genome-scale codon optimization and host range were between 0.64 and 0.84

**Table 1.** List of fungal species analyzed in this work, their host range and genome-scale codon optimization values. Genome-scale codon optimization was calculated using tRNA adaptation indiced (S), codon adaptation indices ($S_{CAI}$) or self-consistent normalized relative codon adaptation indices ($S_{scnRCA}$).

| Species | Host range | Class[*] | S | $S_{CAI}$ | $S_{scnRCA}$ |
|---|---|---|---|---|---|
| *Cryptococcus neoformans* | 800 | Gen. | 0.843 | 0.870 | 0.860 |
| *Rhizoctonia solani* | 690 | Gen. | 0.432 | 0.624 | 0.583 |
| *Botrytis cinerea* | 556 | Gen. | 0.597 | 0.675 | 0.653 |
| *sclerotinia sclerotiorum* | 332 | Gen. | 0.524 | 0.583 | 0.553 |
| *Beauveria bassiana* | 269 | Gen. | 0.616 | 0.619 | 0.747 |
| *Metarhizium acridum* | 228 | Gen. | 0.647 | 0.595 | 0.712 |
| *Aspergillus fumigatus* | 175 | Gen. | 0.609 | 0.643 | 0.684 |
| *Batrachochytrium dendrobatidis* | 153 | Gen. | 0.545 | 0.521 | 0.602 |
| *Verticilium dahliae* | 78 | Gen. | 0.537 | 0.426 | 0.627 |
| *Fusarium graminearum* | 72 | Gen. | 0.788 | 0.818 | 0.819 |
| *Colletotrichum graminicola* | 59 | Gen. | 0.572 | 0.444 | 0.644 |
| *Rhizopus oryzae* | 28 | Gen. | 0.625 | 0.563 | 0.691 |
| *Penicillium digitatum* | 17 | Gen. | 0.545 | 0.602 | 0.630 |
| *Alternaria brassicicola* | 16 | Gen. | 0.400 | 0.592 | 0.628 |
| *Pyrenophora tritici-repentis* | 11 | Gen. | 0.419 | 0.569 | 0.570 |
| *Pseudogymnoascus destructans* | 8 | | 0.484 | 0.513 | 0.505 |
| *Encephalitozoon intestinalis* | 7 | | 0.313 | 0.558 | 0.528 |
| *Melampsora larici-populina* | 7 | | 0.194 | 0.142 | 0.055 |
| *Stagonospora nodorum* | 7 | | 0.155 | 0.431 | 0.462 |
| *Colletotrichum higginsianum* | 6 | | 0.393 | 0.295 | 0.480 |
| *Sporisorium reilianum* | 5 | | 0.476 | 0.200 | 0.429 |
| *Taphrina deformans* | 4 | | 0.403 | 0.679 | 0.671 |
| *Magnaporthe oryzae* | 4 | | 0.437 | 0.424 | 0.575 |
| *Puccinia graminis* | 2 | Spe. | −0.032 | 0.344 | 0.249 |
| *Wolfiporia cocos* | 2 | Spe. | 0.217 | 0.266 | 0.316 |
| *Moniliophthora roreri* | 2 | Spe. | 0.406 | 0.530 | 0.551 |
| *Passalora fulva* | 2 | Spe. | −0.011 | 0.513 | 0.514 |
| *Rozella allomycis* | 1 | Spe. | 0.236 | 0.096 | 0.259 |
| *Nosema ceranae* | 1 | Spe. | 0.021 | −0.155 | −0.085 |
| *Puccinia triticina* | 1 | Spe. | 0.204 | 0.501 | 0.472 |
| *Dothistroma septosporum* | 1 | Spe. | 0.211 | 0.447 | 0.354 |
| *Pseudocercospora fijiensis* | 1 | Spe. | 0.227 | 0.495 | 0.490 |
| *Zymoseptoria tritici* | 1 | Spe. | −0.019 | 0.355 | 0.248 |
| *Blumeria graminis* | 1 | Spe. | 0.116 | −0.187 | 0.164 |
| *Erysiphe necator* | 1 | Spe. | 0.174 | −0.092 | 0.117 |
| *Ophiocordyceps unilateralis* | 1 | Spe. | 0.166 | 0.268 | 0.458 |
| *Gonapodya prolifera* | 0 | np | 0.115 | 0.311 | 0.363 |
| *Rhodotorula toruloides* | 0 | np | 0.344 | 0.385 | 0.470 |
| *Serpula lacrymans* | 0 | np | −0.001 | 0.379 | 0.395 |
| *Laccaria bicolor* | 0 | np | −0.026 | 0.237 | 0.155 |
| *Agaricus bisporus* | 0 | np | 0.361 | 0.432 | 0.400 |

*Table 1 continued on next page*

Table 1 continued

| Species | Host range | Class[*] | S | $S_{CAI}$ | $S_{scnRCA}$ |
|---|---|---|---|---|---|
| *Tuber melanosporum* | 0 | np | 0.230 | 0.319 | 0.349 |
| *Oidiodendron maius* | 0 | np | 0.147 | 0.442 | 0.388 |
| *Myceliophthora thermophila* | 0 | np | 0.246 | 0.123 | 0.323 |
| *Chaetomium globosum* | 0 | np | 0.148 | 0.204 | 0.342 |

[*]The class column indicates whether species were considered as generalist (Gen.), specialist (Spe.) or non-parasitic (np).

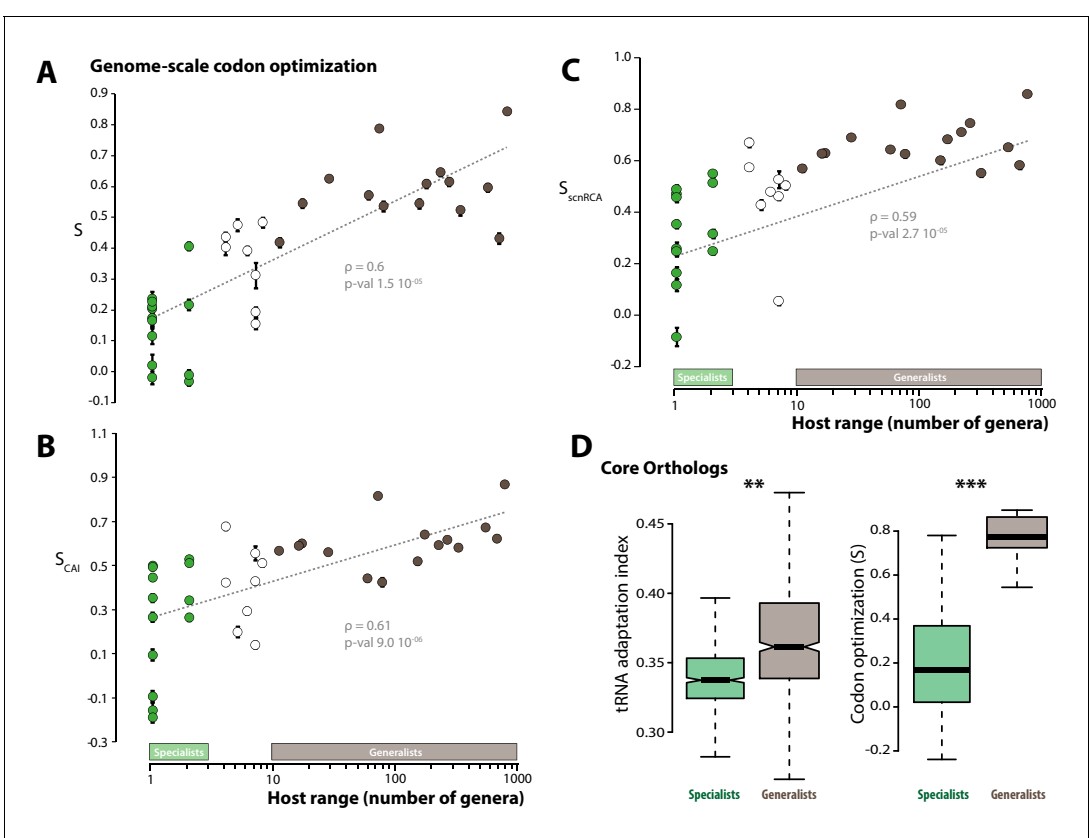

**Figure 2.** Codon optimization correlates with host range in fungal parasites. Genome-scale codon optimization correlates with host range in 36 parasites across the kingdom Fungi. Species considered as specialists (less than four host genera) are shown in green, species considered as generalists (over 10 host genera) are shown in brown. Error bars show 95% confidence interval, dotted line shows logarithmic regression of the data. Codon optimization calculated based on knowledge on the tRNA pool (A), on codon usage in ribosomal protein genes (B) and on self-consistent relative codon adaptation (C) correlated with host range at Spearman $\rho \geq 0.59$ ($p \leq 2.7 \ 10^{-05}$) under phylogenetic independent contrasts. (D) Codon optimization is stronger in core orthologs from generalist fungal parasites than in core orthologs from specialist fungal parasites. Left: Distribution of tRNA adaptation indices in 1620 core ortholog genes show significantly higher values in generalist fungi (**, Welch's t-test p<0.01). Right: Codon optimization calculated as the degree of coadaptation of core ortholog genes to the genomic tRNA pool is significantly higher in generalist fungi (***, Welch's t-test $p<10^{-04}$).

The following source data is available for figure 2:

**Source data 1.** Codon optimization is dependent on breadth of host range but not genome assembly parameters.

**Source data 2.** List of core ortholog genes and their codon adaptation indices.

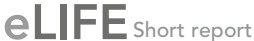

**Figure 3.** Codon optimization and host range co-evolved multiple times across fungal phylogeny. (**A**) Phylogeny, genome-scale codon optimization and host range in 36 parasites across the kingdom Fungi. Nine non-pathogenic species belonging to the major branches of Fungi are shown for comparison. The phylogenetic tree was generated using the TimeTree database (**Hedges et al., 2015**) and PATHd8 (**Britton et al., 2007**). Codon optimization shown as the size of terminal nodes corresponds to the degree S of coadaptation of all genes to the genomic tRNA pool (**dos Reis et al., 2004**). Terminal nodes are sized according to genome-scale codon optimization and colored according to host range (grey was used for non pathogen species). Internal nodes are sized according to reconstructed ancestral genome-scale codon optimization, with ancestral S value indicated as a blue label. (**B**) Correlogram of genome-scale codon optimization and phylogenetic distance along the tree shown in **A**. Dotted lines delimit 95% confidence interval.

The following source data is available for figure 3:

**Source data 1.** Overview of host range features for the 45 fungal species analyzed in this work.
**Source data 2.** Phylogenetic tree of the 45 fungal species analyzed in this work.

for S, $S_{CAI}$ and $S_{scnRCA}$ at the Phylum level in Basidiomycetes and Ascomycetes, and across phyla in non-Dikarya fungi. To unambiguously exclude the impact of phylogeny on the correlation between host range and codon optimization, we calculated Blomberg's K and Pagel's λ as two quantitative measures of phylogenetic signal in genome scale codon optimization along the fungal tree (**Pagel, 1999**; **Blomberg et al., 2003**). We obtained Blomberg's K = 0.28 and Pagel's λ = 0.62. Both measures indicated that evolution of codon optimization is not correlated with the phylogeny along the tree at p<0.01. Then, we used phylosignal (**Keck et al., 2016**) to build a correlogram between phylogenetic distance and codon optimization (**Figure 3B**). Correlations were insignificant (in the range −0.05 to 0.1) along the whole tree. These results support the view that codon optimization is associated with host range in fungal parasites.

## Biased SNP patterns underpin with codon optimization in the generalist parasite *Sclerotinia sclerotiorum*

To get insights into the molecular bases of codon optimization in generalists, we searched for patterns of genome evolution associated with codon optimization at the species level in natural populations of generalist parasites. To this end, we generated genome-wide SNPs data by sequencing five field isolates of the plant pathogenic fungus *Sclerotinia sclerotiorum* (24,344 coding SNPs). *S. sclerotiorum* is the causal agent of *Sclerotinia* rot disease, is notorious for its broad host range encompassing several hundreds of plant genera, and shows strong signatures of codon optimization (S = 0.52). As expected under strong codon optimization, tRNA concentrations determined by small RNA sequencing strongly correlated with the number of tRNA genomic copies, tRNA adaptive values (calculated according to [*dos Reis et al., 2004*]) and codon usage in *S. sclerotiorum* (*Figure 4—figure supplement 1*). We used 131,138 genome-wide coding SNPs of nine isolates of the host-specific plant pathogenic fungus *Zymoseptoria tritici* (*Croll et al., 2013*) as a reference species with weak codon optimization (S = −0.02). Overall, the SNP frequency was higher in the *Z. tritici* population (1.196 SNP.Kb$^{-1}$.isolate$^{-1}$) compared to *S. sclerotiorum* (0.416 SNP.kb$^{-1}$.isolate$^{-1}$). To search for signatures of codon optimization in these populations, we determined the frequency of variants for each codon type. We compared frequency of codon variants to the number of cognate tRNA genomic copies, allowing for wobble pairing (*Crick, 1966*). Frequency of codon variants negatively correlated with the number of tRNA genomic copies in *S. sclerotiorum* (Pearson = −0.60, p = 4.6 10$^{-07}$) but not in *Z. tritici* (Pearson = 0.06, p = 0.62) (*Figure 4A*, *Figure 4—source data 1*).

To further document the molecular bases of codon optimization at the species level, we compared SNP patterns in optimal, intermediate, and non-optimal codons for each amino acid in *S. sclerotiorum* and *Z. tritici* populations. In the *Z. tritici* population, SNP frequencies were not significantly different between optimal and non-optimal codons. By contrast, synonymous SNP frequencies were on average ~1.7-fold lower in optimal codons than in non-optimal codons in *S. sclerotiorum* (Welch's t-test p=1.3 10$^{-03}$) (*Figure 4B*). These findings identify biased synonymous substitutions as a link between generalism and codon optimization. This analysis is independent of codon usage indices and shows that selection for average performances on multiple hosts is reflected in global trends of genome evolution.

## Natural selection drives codon optimization in generalist fungal parasites

Codon bias in a genome can be selectively neutral due to non-randomness in the mutation process. For instance, the identity of favored codons tracks the GC content of the genomes (*Chen et al., 2004*; *Hershberg and Petrov, 2009*). Although *S. sclerotiorum* genome is AT-rich (41.8% GC), synonymous SNPs showed no bias toward AT conversion, suggesting that synonymous SNPs deviated from neutral patterns (*Figure 4—figure supplement 2*). Furthermore, adjusted variant frequencies in *S. sclerotiorum* were similar in intergenic nucleotide triplets and in non-optimal codons but significantly lower in optimal codons (~1.53-fold lower, Welch's t-test p= 1.0 10$^{-11}$) (*Figure 4B*), suggesting that synonymous SNPs in optimal codons may be counter-selected in this species, providing evidence of purifying selection acting on optimal codons. To unambiguously demonstrate that observed SNP patterns lead to codon optimization in *S. sclerotiorum* and are unlikely to result from neutral processes, we simulated the evolution of optimal codon frequencies in *S. sclerotiorum* and *Z. tritici* genomes over 1000 generations (*Figure 4C*). In these simulations, we used either random SNP patterns, or the SNP patterns determined experimentally (*Figure 4—source data 2*). This demonstrated that the codon SNP patterns determined experimentally converge toward increased codon optimality in *S. sclerotiorum* genome, whereas SNP patterns observed in *Z. tritici* genome and random mutation patterns do not. Thus, patterns of evolution toward increased codon optimality were detected in *S. sclerotiorum* but not in *Z. tritici* populations, and deviate significantly from neutral evolution.

To estimate the likelihood that observed codon biases result from selective rather than neutral processes in other fungal genomes, we used permutation tests on the number of genomic tRNA copies to calculate probability distribution of S under no selection (*dos Reis et al., 2004*). A total of 14 species showed signatures of selection for codon optimization with a p<0.1, among which one was a specialist (*M. roreri*) and one had eight host genera (*P. destructans*). Twelve out of 15 (80%)

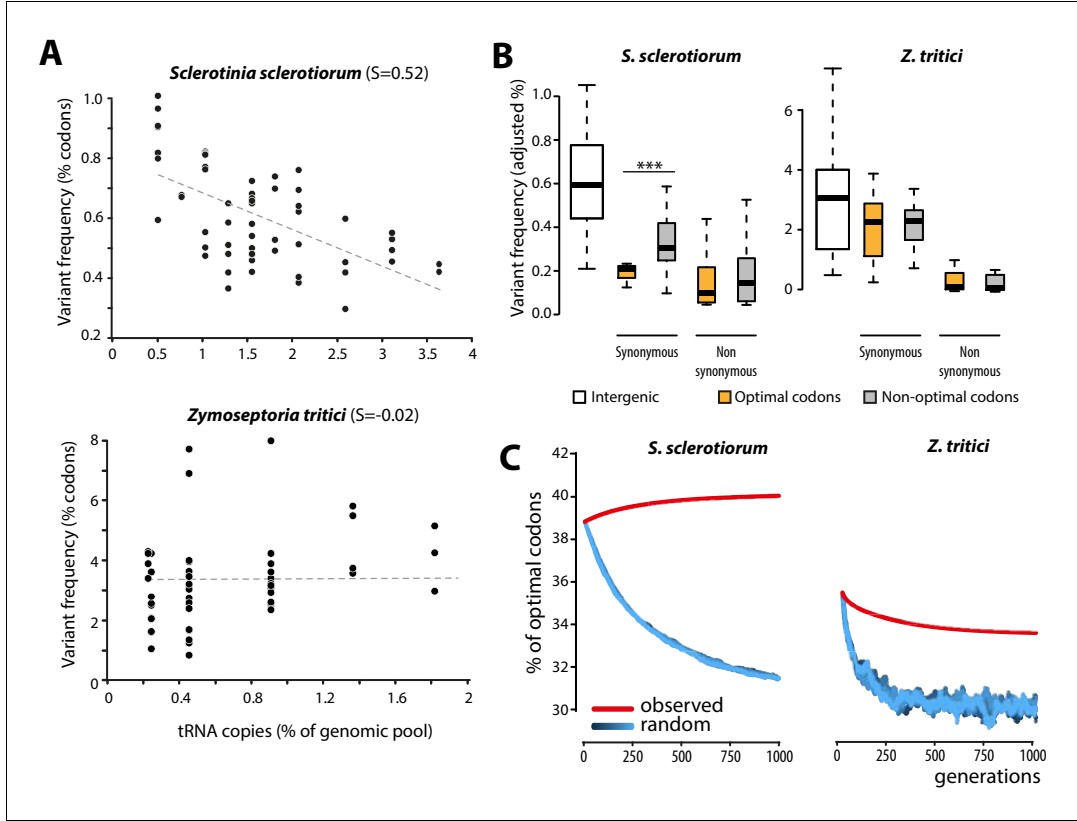

**Figure 4.** Biased synonymous substitution patterns underpin codon optimization in local populations of a generalist but not a specialist fungal parasite. (**A**) Genome-wide frequencies of variant codons in local populations of the host generalist *Sclerotinia sclerotiorum* and the host specialist *Zymoseptoria tritici*, according to the number of genomic copies of cognate tRNAs. The number of cognate tRNAs for each codon type was determined using wobble rules for codon-anticodon pairing. Dotted lines show linear regression of the data (*Z. tritici*: Pearson ρ = 0.06; p=0.62; *S. sclerotiorum* ρ = −0.60; p=4.6 $10^{-07}$). (**B**) Adjusted variant frequencies for intergenic nucleotide triplets, optimal and non-optimal codons. Synonymous and non-synonymous SNPs are shown separately. Differences between optimal and non-optimal codon rates were assessed by Welch's t-test (\*\*\*p<0.001). (**C**) Predicted evolution of genome-wide content in optimal codons in *S. sclerotiorum* and *Z. tritici* based on observed and random mutation patterns.

The following source data, source code and figure supplements are available for figure :

**Source data 1.** Codon statistics for *S. sclerotiorum* and *Z. tritici* genomes.
**Source data 2.** Frequency of codon substitutions in *S. sclerotiorum* and *Z. tritici* populations (as % of all codons).
**Source code 1.** Python scripts for in silico evolution of codon usage.
**Figure supplement 1.** Experimental determination of *S. sclerotiorum* tRNA accumulation supports a good correlation between genomic copy numbers and tRNA accumulation.
**Figure supplement 2.** Analysis of Single Nucleotide Polymorphisms (SNPs) in a natural population of the generalist plant pathogen *Sclerotinia sclerotiorum*.

generalist species showed selection for codon optimization with a p<0.1, confirming that adaptive translation is stronger in generalist species. We conclude that natural selection on protein translation efficiency or accuracy drives evolution toward codon optimization in generalist parasites through

biased synonymous SNPs. These evolutionary patterns are consistent with an improvement of average performances on multiple hosts for generalist parasites.

## Codon optimization signatures associate with host colonization

To further support the link between codon optimization and the ability to colonize multiple hosts, we analyzed signatures of codon optimization in host-induced genes of 15 fungal parasite species from multiple families of the Ascomycetes and Basidiomycetes (*Figure 5—source data 1*) (*Amselem et al., 2011*; *Duplessis et al., 2011a*, *2011b*; *Gao et al., 2011*; *Morton et al., 2011*; *O'Connell et al., 2012*; *Xiao et al., 2012*; *Zhang et al., 2012*; *Hacquard et al., 2013*; *Bailey et al., 2014*; *Chen et al., 2014*; *Kellner et al., 2014*; *de Bekker et al., 2015*; *Bradshaw et al., 2016*). We found that host-induced genes were enriched ~3.1-fold among the top 10% codon-adapted genes in generalist parasites but not in specialists. Overall, genes induced during host colonization distributed predominantly among genes with high codon adaptation indices in generalist but not in specialist parasites (*Figure 5A*, *Figure 5—source data 2*). Parasites use secreted proteins to facilitate infection of their hosts. We therefore analyzed codon optimization in secretome genes of 45 fungal species (*Figure 5—source data 3*). We found that secretome genes were enriched over 1.6-fold among the top 10% codon-adapted genes in generalist parasites but only ~1.2-fold in specialists and non-parasitic fungi. Overall, predicted secretome genes distributed predominantly among genes with high codon adaptation indices in generalist parasites but not in specialist and non parasitic fungi (*Figure 5B*). We calculated S in each genome for secreted and non-secreted protein genes (*Figure 5C*, *Figure 5—source data 4*). In our genome set, non-parasitic fungi had similar S for secreted and non-secreted proteins. The average S for secreted proteins (0.212) was slightly higher than for non-secreted proteins (0.137) in specialists (Student's paired t-test p=0.054). In generalists, the average S for secreted proteins (0.661) was significantly higher than for non secreted proteins (0.558, Student's paired t-test p=1.12 $10^{-03}$). Increased codon optimization in generalists was not only clear for genes encoding secreted proteins and host-induced genes, but also across their entire genome ('other genes' in *Figure 5C*). To get a global overview of cellular functions enriched in codon optimized genes in generalists, we analyzed normalized tRNA adaptation indices for genes annotated with Gene Ontologies (GO) in generalist and specialist genomes (*Figure 5D*, *Figure 5—source data 5*). Together with secreted enzymes, GOs related to translation and central metabolism showed higher codon optimization in generalists than in specialists. Conversely, GOs related to transcription, transposable elements and phosphorelay signal transduction were better optimized in specialists than in generalists. These results suggest that high codon optimization is an adaptation to host colonization and particularly to the colonization of multiple hosts.

## Discussion

Our study reveals that codon optimization through biased synonymous substitutions is a common feature in the evolution of generalist parasites and is associated with the colonization of multiple hosts. Because host colonization by generalist often requires diverse and relatively complex secreted proteins, these organisms are expected to require higher protein translation efficiency to compete with co-occuring specialist microbes. In agreement with this model, we found that natural selection for codon optimization correlates with host range in fungal parasites. We describe patterns of purifying selection acting on optimal codons in generalist parasites, providing a molecular basis for genome scale adaptation to the colonization of multiple hosts.

The increasing amount of genome sequences available for specialist fungal parasites revealed massive losses of enzymes associated with the ability to thrive and reproduce on living hosts (*Schirawski et al., 2010*; *Spanu et al., 2010*). Specialists rely largely on SSPs subverting host cell functions. Functional analyses have shown that SSPs from filamentous pathogens can function as effectors, facilitating the colonization of susceptible hosts while triggering resistance in some host genotypes by activating resistance (R) proteins (*Dodds and Rathjen, 2010*). In specialist species, the one-to-one relationship, either direct or indirect, between effectors and R proteins favors evolution through non-synonymous mutations, to increase effector diversity and escape R protein recognition (*Raffaele and Kamoun, 2012*). By contrast, pathogen populations exposed to weaker host selection, caused by quantitative resistance genes, or disruptive selection due to heterogeneity in the host population, are expected to evolve adapted genetic variants less rapidly (*McDonald and Linde,*

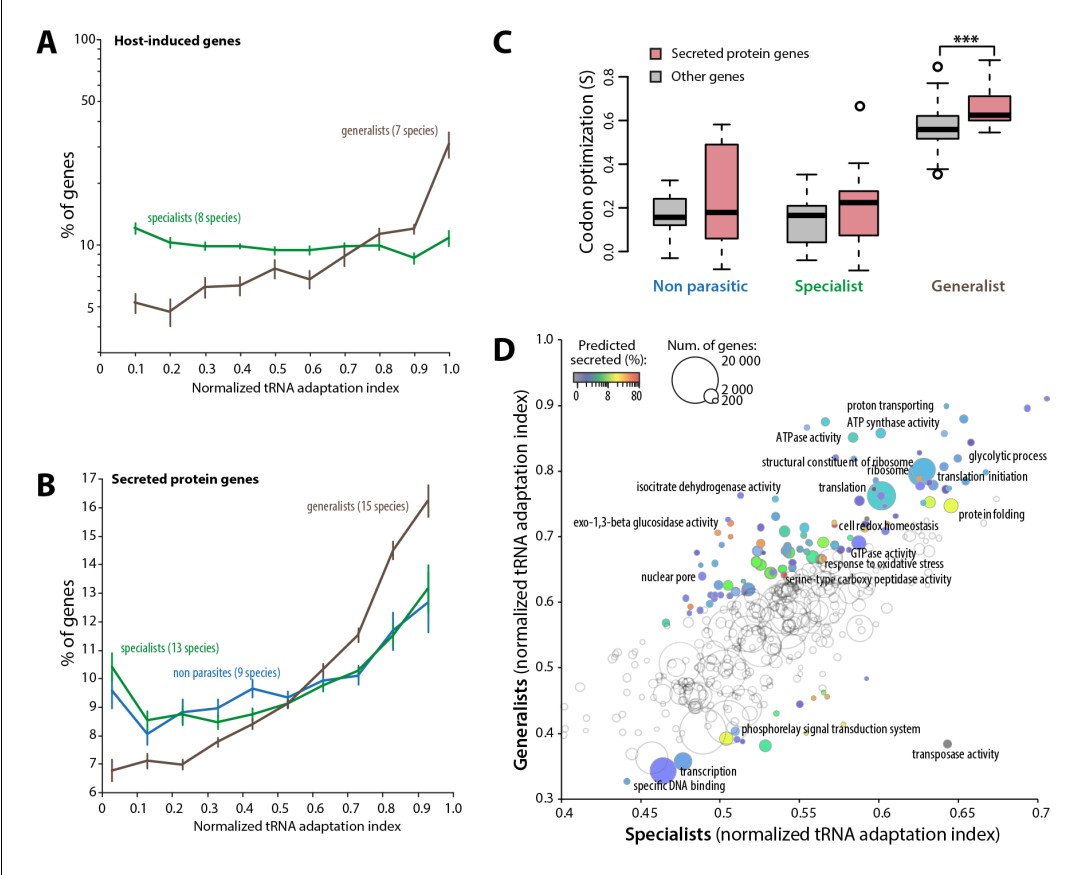

**Figure 5.** Codon optimization strongly associates with host colonization in generalist fungal parasites. (**A**) Genes induced during host infection are enriched among high tRNA adaptation index genes in generalist but not specialist parasite genomes. Error bars show standard error of the mean. (**B**) Genes encoding predicted secreted proteins are more strongly enriched among high tRNA adaptation index genes in generalist parasite genomes compared to specialist and non-parasitic fungi genomes. Error bars show standard error of the mean. (**C**) Degree of coadaptation of secreted protein genes and other genes to the genomic tRNA pool in non parasitic, specialist and generalist fungi (*** Student's paired t-test p<0.002). (**D**) Distribution of tRNA adaptation indices according to gene functions in generalist and specialist fungal parasites. For each Gene Ontology (GO), the average normalized tRNA adaptation indices in all genes from generalist and specialist genomes are shown. Bubbles are sized according to the total number of genes, and colored according to the percentage of predicted secreted proteins when values for generalists and specialists differ by over 10%. Selected GOs are labeled.

The following source data is available for figure 5:

**Source data 1.** Summary of gene expression data used for the analysis of tAI in host-induced genes.
**Source data 2.** Distribution of host-induced genes according to tAI (as % of all host-induced genes).
**Source data 3.** Distribution of secreted protein genes according to tAI (as % of all host-induced genes).
**Source data 4.** Codon optimization values in secreted and non-secreted proteins for each of the 45 fungal genomes analyzed in this work.
**Source data 5.** Distribution of tRNA adaptation indices per Gene Ontology in generalist and specialist genomes.

*2002*; *Roux et al., 2014*). Our analyses indeed support the view that selection driven by multiple hosts is associated with relatively low rates of evolution of protein variants but instead associates with the directional evolution of synonymous genetic variants. Translational selection may, nevertheless, be active in the genome of specialists but on limited gene sets only, resulting in a low signal in genome-scale analyses (*dos Reis et al., 2004*).

We noted a trend toward longer proteins encoded in the genome of generalist fungi, notably secreted proteins, consistent with the abundance of SSPs in specialists (*Spanu et al., 2010*) and previous comparative analyses (*Kim et al., 2016*). We associated longer proteomes in generalists with higher selection on codon optimization at the whole genome scale. The relative strength of translational selection on individual genes may, however, vary significantly within a given genome. Gene length increases in relation to evolutionary age, with conserved genes under purifying selection generally being longer and harboring higher frequency of optimal codons (*Prat et al., 2009*; *Wolf et al., 2009*). Neofunctionalization in duplicated genes has also been associated with increased gene length (*He and Zhang, 2005*). Duplications frequently underlie domain loss and gain in eukaryotic proteins, which is an important mechanism for the evolution of new functions (*Buljan et al., 2010*; *Peisajovich et al., 2010*). Natural selection may promote protein domains recombination to increase the versatility of fungal proteins functions, and thereby contribute to host range expansion.

The adaptive evolution of effectors that increase the colonization of one host species also increases the risk of detection in other host species by triggering their immune system. Furthermore, natural selection that maximizes parasite fitness on one particular host species might result in decreased fitness on an alternative host, leading to specialization (reduced host range) or host shift (*Dong et al., 2014*). The evolution of generalism has been associated to hybridization events (*McMullan et al., 2015*; *Menardo et al., 2016*), horizontal transfer of genetic elements (*Ma et al., 2010*; *Hu et al., 2014*), and an overall increase in the repertoire of protein-coding genes (*Hu et al., 2014*). This observation is consistent with the prominent role in pathogenic success of large repertoires of host-degrading enzymes encoded by the genome of generalist parasites (*King et al., 2011*; *Zhao et al., 2013*). Catalytic efficiency of these enzymes is crucial to perform in host environment. Although the catalytic activity of enzymes is typically modulated by adaptation of protein sequences, high reaction rates for efficient host colonization may be reached by modulating the amount of protein synthesized during colonization. However, an increase in proteins production may lead to a metabolic or structural burden of the translation machinery. For instance, the massive production of virulence factors by the generalist bacterial pathogen *Ralstonia solanacearum* reduces its metabolic versatility and thus its host colonization capacity (*Perrier et al., 2016*; *Peyraud et al., 2016*). Natural selection for the optimization of protein translation, through ribosome pausing time, translation error rates and co-translational protein folding, has been widely associated with codon usage biases (*Drummond and Wilke, 2008*; *Tuller et al., 2010*; *Shah and Gilchrist, 2011*). The adaptive evolution of the translation machinery described here is predicted to favor generalists' performances in heterogeneous host populations, by reducing the cost of large and diverse sets of secreted enzymes production on proliferation. The mitigation of the tradeoff between cell proliferation and protein synthesis (*Li et al., 2014*) by codon optimization could explain how generalist lineages have maintained or gained the ability to produce diverse sets of secreted enzymes, allowing the colonization of genetically diverse hosts. Codon optimization may affect the expression of effectors and other virulence factors, modifying the degree of infection or enabling the colonization of new hosts opposing quantitative disease resistance mechanisms. These findings reveal that parasite host range variation is a prime example of an adaptive phenotype related to codon optimality and translational regulation.

## Materials and methods

### Simulation of codon optimization effect on protein biosynthesis

We designed a mathematical model of the cell translation machinery in order to undertake the analysis of the impact of codon optimization parameters on the cell physiology (*Figure 1—source data 1*). The model describes the variation of intracellular concentration, P(i), of three proteins sets (i): the free ribosomal proteins P(r), the proteins destined to be secreted P(s) and intracellular non-secreted proteins P(n), depending on their synthesis rate and the dilution rate due to cell growth. Crowding effect (*Beg et al., 2007*), that is the maximal number of proteins that a finite cell can contain, was modeled by adding a limitation function to the elongation rate which is asymptotic to the maximal proteins content. The ordinary differential equations were solved numerically using the COPASI software (RRID:SCR_014260) (*Hoops et al., 2006*) until a steady-state was reached. mRNA(*i*) concentrations were assumed to be stable with no dilution due to cell growth. The initiation constants

between mRNA and ribosomes were set to a non-limiting value, similar for the three kinds of proteins sets (r, n and s). A stoichiometric constrain of 79 ribosomal proteins bound per mRNA(i) was added in the COPASI model. The initial parameters used for the different analysis are listed in *Figure 1—source data 1*. These parameters were set to be in the physiological range of a cell accordingly to data from *Bremer and Dennis (2008)*. The concentrations of the biomolecules in the model are in mmol•ml$^{-1}$. The maximal growth rate was defined as the dilution rate (μ) above which the rate of the proteins biosynthesis is lower than the dilution rate, leading to the collapse of the proteins concentration. We considered situations in which 14.5% of total mRNAs corresponded to mRNAs for secreted proteins and 12% corresponded to mRNAs for ribosomal proteins, and analyzed the impact of varying codon decoding rate on protein production under our cellular model.

## Codon optimization across the kingdom fungi

Genome assemblies and CDS files for 45 fungal species were downloaded from the repositories given in *Figure 2—source data 1*. For every species, the repertoire of tRNA genes was determined using the tRNAscan-SE 1.21 server (RRID:SCR_010835) (*Schattner et al., 2005*) with the 'Source' parameter set as 'Eukaryotic' and 'Search Mode' set as 'Default'. For 42 species, only predicted full length CDS were considered for codon optimization analysis. For this, CDS lacking a start and a stop codon were filtered out. For *Puccinia graminis*, *Serpula lacrymans* and *Colletotrichum higginsianum*, since >15% of CDS lacked a predicted stop codon in these genomes, we filtered out CDS lacking a start codon only. tRNA adaptation indices (tAIs) were calculated for a total of 569,744 genes using the get.tai function in R (*dos Reis et al., 2004*) with the 'sking' parameter set to '0' (Eukaryote) for the get.ws function. The degree of codon optimization in each genome was determined using the get.s function in R (*dos Reis et al., 2004*). Confidence intervals were determined using the CI function from the package 'psychometric' in R. For each genome, the probability of selection acting on codon optimization was determined using the ts.test function in R (*dos Reis et al., 2004*) with sample size = 500 and n = 1000 and p values were adjusted using Bonferroni correction. The repertoire of ribosomal proteins was identified based on interproscan annotation or de novo gene annotation performed with Blast2GO version 3.3 (RRID:SCR_005828) (*Conesa et al., 2005*). For each genome, the CDS of full length ribosomal proteins were used to compute a codon usage table with the CUSP function in EMBOSS (RRID:SCR_008493) (*Rice et al., 2000*) to calculate codon adaptation indices (CAI) for every gene in the corresponding genome, using the CAI function in EMBOSS. The degree of codon optimization at the whole genome scale was determined using the get.s function in R with gene CAI or scnRCA instead of tAI. Host range was determined according to sources listed in *Figure 2—source data 2*. For most plant pathogens, the Systematic Mycology and Microbiology Laboratory Fungus-Host Database of the United States Department of Agriculture was used (*Farr and Rossman, 2016*). For *Cryptococcus neoformans*, *Aspergillus fumigatus*, *Beauveria bassiana*, and *Metarhizium acridum*, a list of infected Orders were retrieved. We collected the list of Genera recorded in these Orders through various sources (*Figure 2—source data 2*). The likelihood that a parasite can infect two host species decreases continuously with phylogenetic distance between the hosts (*Gilbert and Webb, 2007*). In their study, Gilbert and Webb reported that the proportion of hosts colonized was about ~20% at a 300 Mya phylogenetic distance, decreasing logarithmically. We considered that 8% of Genera in a given Order were likely to be host for the above-mentioned pathogens to reach a conservative estimate. Core orthologous genes were identified through a BlastP search against a database of 36 complete fungal proteomes, using the CEGMA set downloaded from http://korflab.ucdavis.edu as a query. Only orthologous groups containing a single hit in all fungal species were selected for further analysis. tRNA adaptation indices for each genome were median-normalized before analyzing their distribution in COGs. We used the TimeTree database (*Hedges et al., 2015*) to obtain ages for a maximum of nodes of the tree (calibration points). Then, we ultrametricized the tree using PATHd8 (*Britton et al., 2007*). We tested for a potential presence of a phylogenetic signal estimated by Blomberg's K, Pagel's λ and autocorrelogram with the phylosignal R-package (*Keck et al., 2016*). We calculated phylogenetic independent contrasts (*Felsenstein, 1985*) and verified the significance of trait correlations using the pic function from the ape package in R.

## Sequencing of *S. sclerotiorum* isolates

Isolates C014, P163, P314 and C104 were provided by Bruno Grezes-Besset (Biogemma, Mondon-ville, France) and were collected from infected rapeseed fields near Blois (France) in 2010. Isolate FrB5, collected from clover seed lot in Dijon (France) in 2010, was obtained from *Vleugels et al. (2013)*. Isolate 1980 (ATCC18683) was collected in prior 1970 on bean pods near New York (*Maxwell and Lumsden, 1970*) and was provided by Martin Dickman (Texas A and M University, USA). For IGS phylogeny, Intergenic sequences (IGS) were retrieved on sequenced genomes with the Blast tool of CLC Genomics software (RRID:SCR_011853). The sequences were aligned with MAFFT v7 (RRID:SCR_011811) (*Katoh and Standley, 2013*), and the phylogenetic tree was build with PhyML 3 (*Guindon et al., 2010*) with SH-like approximate likelihood ratio test (aLRT) as support statistics. The tree was mid-point rooted using MEGA6 software (RRID:SCR_000667) (*Tamura et al., 2013*). *Sclerotinia sclerotiorum* isolates were grown in liquid potato dextrose medium (P6685 - Sigma) for 4 days at 24°C under shaking at 180 rpm. Cultures were filtered using vacuum and a frit-ted glass column protected with a doubled miracloth membrane and ground in liquid nitrogen. One gram of mycelium powder was used for DNA extraction using a DNA Maxi Kit (Quiagen) and follow-ing manufacturer's instructions. DNA sequencing was outsourced to Fasteris SA (Switzerland) to pro-duce Illumina paired-end reads (2 × 100 bp) using a HiSeq 2500 instrument. Paired-end reads for each *S. sclerotiorum* isolate were mapped to version 2 of the *S. sclerotiorum* 1980 reference strain genome (*Derbyshire et al., 2017*) using the mapping function of the CLC Genomics software. Vari-ant call was performed on mapped reads with the Fixed Ploidy Variant Detection function of the CLC Genomics software. The resulting total variants for each strain were further filtered for homozy-gous single nucleotide variants.

## Analysis of codon polymorphisms in plant pathogen populations

SNPs in gene coding sequences were extracted and mapped to the corresponding codons using R scripts to calculate the percentage of polymorphic codons of each type. The number of tRNAs able to decode each codon type is given as a percentage of the genomic tRNA pool that was determined based on tRNA gene predictions obtained through tRNAscan-SE. Whenever isoacceptor codons had no cognate tRNA predicted, the number of tRNA copies was divided evenly among isoacceptor codons according to general codon-anticodon wobble rules as described in *dos Reis et al. (2004)*. We considered that the 11 codons that did not have isodecoder tRNAs in *Zymoseptoria tritici* were decoded evenly by the four tRNA genes that had no accepting codons identified by tRNAscan-SE. Codons with the highest percentage of decoding tRNAs were considered as optimal. In cases where the highest percentage of decoding tRNA was shared among several synonymous codons, codons having the highest number of cognate tRNA copies in the genome or codons with the highest usage in the genome were optimal. Codons with the lowest percentage of decoding tRNAs were consid-ered as non-optimal. In cases where the lowest percentage of decoding tRNA was shared among several synonymous codons, codons with the lowest usage in the genome were considered non-opti-mal. Other codons were considered as intermediate.

Variant frequencies (in % of codons) were calculated by counting the number of codons harboring a SNP divided by the total number of this codon type in the genome. These raw variant frequencies were used in simulation of the evolution of codon optimality. A random base change within a codon is more likely to result in a nonsynonymous or stop-gain mutation than a synonymous mutation, hence we expect the frequency of non-synonymous codon variants to be higher than the frequency of synonymous codon variants under neutral selection. Therefore, to compare synonymous and non-synonymous variant frequencies, we adjusted non-synonymous variant frequencies dividing them by their likelihood to occur if changes were random. Variant frequencies in intergenic nucleotide triplets were used as an estimate for neutral patterns of evolution. For this, we extracted for each intergenic SNP the corresponding triplets. Variant frequencies (% of nucleotide triplets) were calculated by counting the number of triplets of a given type harboring a SNP divided by the total number of this triplet type in intergenic regions of the genome, determined using the COMPSEQ function in EMBOSS. Frequencies of variants in intergenic triplets were adjusted to compare to synonymous and non-synonymous variant frequencies. For this, we use as a correction factor the ratio between the adjusted total codon variant rate (synonymous variant rate + adjusted non-synonymous variant rate) over the unadjusted total codon variant rate (synonymous variant rate + non-synonymous

variant rate). SNP frequencies in genes (in Kbp$^{-1}$) were calculated counting the number of SNPs per genes and divided by the length of gene coding sequence in Kbp.

## Determination of tRNA genes expression

Small RNAs were obtained from samples of *Arabidopsis thaliana* Col-0 ecotype plants infected by *S. sclerotiotum* strain 1980 and samples of *S. Sclerotiorum* strain 1980 grown in Potato Dextrose Broth as previously described. Small RNA were prepared using the NucleoSpin miRNA kit (Macherey-Nagel) following instructions of the manufacturer. Small RNAs sequencing was outsourced to Fasteris SA (Switzerland) to produce Illumina reads using a HiSeq 2500 instrument (small RNA libraries with expected insert size from 18 to 30 bp). Reads were imported into CLC Genomics Workbench 8.5 (QUIAGEN), and sequences were mapped on version 2 of the *S. sclerotiorum* strain 1980 genome (*Derbyshire et al., 2017*). Read counts at predicted tRNA loci were obtained using the Small RNA Analysis toolkit of the CLC Genomics Workbench 8.5 software. Only perfect matches to the *S. sclerotiorum* tRNA genomic loci on the correct strand were accepted, and inserts less than 15pb were discarded. Reads collected in planta or in vitro and aligned to the different copies of a given tRNA type in the genome were summed up and compared to tRNA gene copy numbers, codon usage or tRNA adaptive values.

## Simulations of the evolution of codon optimality

Codons were classified as optimal, intermediate and non-optimal as described previously, unadjusted variant frequencies were used. The initial state in simulations corresponded to codon usage measured in *S. sclerotiorum* and *Z. tritici* reference genomes. To simulate the long-term evolution of codon optimality in *S. sclerotiorum* and *Z. tritici*, we developed two python scripts. The 'observed' simulator uses the frequencies of all 64x64 = 4096 possible codon substitutions measured in *S. sclerotiorum* and *Z. tritici* natural populations, kept constant over the simulation. At each evolutionary step, the frequencies of the 64 codons in each genome are mutated according to the codon substitution frequencies table. The total proportion of optimal, intermediate and non-optimal codons in each genome is computed after all measured codon substitutions have been accounted for (designated as one 'generation'). To test if the observed long-term evolution of codon optimality deviated significantly from random mutation patterns, a 'random' simulator was implemented. In these simulations, bulk frequencies of triplets toward synonymous and not synonymous mutations were kept constant, whereas the outcome of mutations was determined randomly at each iteration. The number of mutations at each generation also varied randomly in order to obtain an uneven distribution. Simulations were repeated until an asymptotic behavior was apparent. The Python source code for these two simulators is provided in *Figure 4—source code 1*.

## Codon optimization and genes function

To normalize tAIs across multiple genomes, tAI values were ordered in ascending order in each genome, and their rank given as a percentage (0 for lowest rank corresponding to the lowest tAI in a given genome up to 1 for the highest rank corresponding to the highest tAI in this genome). The source of data and pipelines used to identify genes significantly induced are listed in (*Figure 5— source data 1*). The predicted secreted protein genes were determined using SignalP 4.0 (*Petersen et al., 2011*) with default parameters. For each genome, genes were classified into secreted protein genes (signal peptide identified by SignalP 4.0) and non-secreted protein genes (all other genes). The degree of codon optimization in these two gene subsets were computed for each genome using the get.s function in R (*dos Reis et al., 2004*). The average normalized tAI for all genes annotated with a given Gene Ontology was computed for generalist and specialist genomes. GO mapping was retrieved from the repositories given in *Figure 2—source data 1* or performed de novo with Blast2GO software. Only GOs associated with 200 genes or more were analyzed.

## Acknowledgements

We thank Marielle Barascud, Aline Lacaze and Remy Vincent for technical assistance, Bruno Grezes-Besset and Tim Vleugels for providing *S. sclerotiorum* isolates. This work was supported by a starting grant of the European Research Council (ERC-StG 336808 project VariWhim) to SR and the French Laboratory of Excellence project TULIP (ANR-10-LABX-41; ANR-11-IDEX-0002–02). MD and RPO are

supported by the Australian Grains Research and Development Corporation and Curtin University. Sequences are deposited in GenBank under the BioProject ID PRJNA341340.

## Additional information

### Funding

| Funder | Grant reference number | Author |
|---|---|---|
| European Research Council | ERC-StG 336808 | Thomas Badet<br>Remi Peyraud<br>Malick Mbengue<br>Olivier Navaud<br>Sylvain Raffaele |
| Labex TULIP | ANR-11-IDEX-0002-02 | Thomas Badet<br>Remi Peyraud<br>Malick Mbengue<br>Olivier Navaud<br>Adelin Barbacci<br>Sylvain Raffaele |
| Australian grains research and development corporation | | Mark Derbyshire<br>Richard P Oliver |
| Curtin University of Technology | | Mark Derbyshire<br>Richard P Oliver |
| Labex TULIP | ANR-11-IDEX-0002-02 | Thomas Badet<br>Remi Peyraud<br>Malick Mbengue<br>Olivier Navaud<br>Adelin Barbacci<br>Sylvain Raffaele |

The funders had no role in study design, data collection and interpretation, or the decision to submit the work for publication.

### Author contributions

TB, Data curation, Formal analysis, Investigation, Methodology, Writing—review and editing; RP, Data curation, Formal analysis, Methodology, Writing—original draft, Writing—review and editing; MM, Resources, Formal analysis, Investigation, Methodology, Writing—review and editing; ON, Formal analysis, Methodology, Writing—review and editing; MD, RPO, Resources, Writing—review and editing; AB, Software, Formal analysis, Methodology, Writing—review and editing; SR, Conceptualization, Formal analysis, Supervision, Funding acquisition, Writing—original draft, Project administration, Writing—review and editing

### Author ORCIDs

Adelin Barbacci, http://orcid.org/0000-0003-3156-272X
Sylvain Raffaele, http://orcid.org/0000-0002-2442-9632

## Additional files

### Major datasets

The following dataset was generated:

| Author(s) | Year | Dataset title | Dataset URL | Database, license, and accessibility information |
|---|---|---|---|---|
| Mbengue M, Navaud O, Raffaele S | 2016 | Sclerotinia sclerotiorum isolates genome sequence | https://www.ncbi.nlm.nih.gov/bioproject/342788 | Publicly available at the NCBI BioProject (accession no: PRJNA342788). |

The following previously published datasets were used:

| Author(s) | Year | Dataset title | Dataset URL | Database, license, and accessibility information |
|---|---|---|---|---|
| de Bekker C, Ohm RA, Loreto RG, Sebastian A, Albert I, Merrow M, Brachmann A, Hughes DP | 2015 | Ophiocordyceps unilateralis genome sequence | http://www.ncbi.nlm.nih.gov/nuccore/LAZP00000000.1/ | Publicly available at NCBI Nucleotide (accession no. LAZP00000000.1) |
| James TY, Pelin A, Bonen L, Ahrendt S, Sain D, Corradi N, Stajich JE | 2013 | Rozella allomycis genome sequence | http://genome.jgi.doe.gov/Rozal1_1/Rozal1_1.home.html | Publicly available at genome.jgi.doe.gov |
| Ma LJ, Ibrahim AS, Skory C, Grabherr MG, Burger G, Butler M, Elias M, Idnurm A, Lang BF, Sone T, Abe A, Calvo SE, Corrochano LM, Engels R, Fu J, Hansberg W, Kim JM, Kodira CD, Koehrsen MJ, Liu B, Miranda-Saavedra D, O'Leary S, Ortiz-Castellanos L, Poulter R, Rodriguez-Romero J, Ruiz-Herrera J, Shen YQ, Zeng Q, Galagan J, Birren BW, Cuomo CA, Wickes BL | 2009 | Rhizopus oryzae genome sequence | http://genome.jgi.doe.gov/Rhior3/Rhior3.home.html | Publicly available at genome.jgi.doe.gov |
| Cornman RS, Chen YP, Schatz MC, Street C, Zhao Y, Desany B, Egholm M, Hutchison S, Pettis JS, Lipkin WI, Evans JD | 2009 | Nosema ceranae genome sequence | http://www.ebi.ac.uk/ena/data/view/GCA_000988165.1 | Publicly available at the EBI European Nucleotide Archive (accession no: GCA_000988165.1). |
| Corradi N, Pombert JF, Farinelli L, Didier ES, Keeling PJ | 2010 | Encephalitozoon intestinalis genome sequence | http://genome.jgi.doe.gov/Encin1/Encin1.home.html | Publicly available at genome.jgi.doe.gov |
| Batrachochytrium dendrobatidis Sequencing Project, Broad Institute of Harvard and MIT | 2016 | Batrachochytrium dendrobatidis genome sequence | https://www.broadinstitute.org/fungal-genome-initiative/batrachochytrium-genome-project | Publicly available at www.broadinstitute.org |
| Chang Y, Wang S, Sekimoto S, Aerts AL, Choi C, Clum A, LaButti KM, Lindquist EA, Yee Ngan C, Ohm RA, Salamov AA, Grigoriev IV, Spatafora JW, Berbee ML | 2015 | Gonapodya prolifera genome sequence | http://genome.jgi.doe.gov/Ganpr1/Ganpr1.home.html | Publicly available at genome.jgi.doe.gov |
| Schirawski J, Mannhaupt G, Münch K, Brefort T, Schipper K, Doehlemann G, Di Stasio M, Rössel N, Mendoza-Mendoza A, Pester D, Müller O, Winterberg B, Meyer E, Ghareeb H, Wollen- | 2010 | Sporisorium reilianum genome sequence | http://genome.jgi.doe.gov/Spore1/Spore1.home.html | Publicly available at genome.jgi.doe.gov |

| | | | | |
|---|---|---|---|---|
| berg T, Münsterkötter M, Wong P, Walter M, Stukenbrock E, Güldener U, Kahmann R | | | | |
| Zhu Z, Zhang S, Liu H, Shen H, Lin X, Yang F, Zhou YJ, Jin G, Ye M, Zou H, Zhao ZK | 2012 | Rhodotorula toruloides genome sequence | http://genome.jgi.doe.gov/Rhoto1/Rhoto1.home.html | Publicly available at genome.jgi.doe.gov |
| Puccinia Group Sequencing Project, Broad Institute of Harvard and MIT | 2016 | Puccinia triticina genome sequence | https://www.broadinstitute.org/scientific-community/science/projects/fungal-genome-initiative/puccinia-comparative-genomic-projects | Publicly available at www.broadinstitute.org |
| Duplessis S, Cuomo CA, Lin YC, Aerts A, Tisserant E, Veneault-Fourrey C, Joly DL, Hacquard S, Amselem J, Cantarel BL, Chiu R, Coutinho PM, Feau N, Field M, Frey P, Gelhaye E, Goldberg J, Grabherr MG, Kodira CD, Kohler A, Kues U, Lindquist EA, Lucas SM, Mago R, Mauceli E, Morin E, Murat C, Pangilinan JL, Park R, Pearson M, Quesneville H, Rouhier N, Sakthikumar S, Salamov AA, Schmutz J, Selles B, Shapiro H, Tanguay P, Tuskan GA, Henrissat B, Van de Peer Y, Rouzé P, Ellis JG, Dodds PN, Schein JE, Zhong S, Hamelin RC, Grigoriev IV, Szabo LJ, Martin F | 2011 | Puccinia graminis genome sequence | http://genome.jgi.doe.gov/Pucgr2/Pucgr2.home.html | Publicly available at genome.jgi.doe.gov |
| Loftus BJ, Fung E, Roncaglia P, Rowley D, Amedeo P, Bruno D, Vamathevan J, Miranda M, Anderson IJ, Fraser JA, Allen JE, Bosdet IE, Brent MR, Chiu R, Doering TL, Donlin MJ, D'Souza CA, Fox DS, Grinberg V, Fu J, Fukushima M, Haas BJ, Huang JC, Janbon G, Jones SJ, Koo HL, Krzywinski MI, Kwon-Chung JK, Lengeler KB, Maiti R, Marra MA, Marra RE, Mathewson CA, Mitchell TG, Pertea M, Riggs FR, Salzberg SL, Schein JE, Shvartsbeyn A, Shin | 2005 | Cryptococcus neoformans genome sequence | http://genome.jgi.doe.gov/Cryne_JEC21_1/Cryne_JEC21_1.home.html | Publicly available at genome.jgi.doe.gov |

| | | | | |
|---|---|---|---|---|
| H, Shumway M, Specht CA, Suh BB, Tenney A, Utterback TR, Wickes BL, Wortman JR, Wye NH, Kronstad JW, Lodge JK, Heitman J, Davis RW, Fraser CM, Hyman RW | | | | |
| Floudas D, Binder M, Riley R, Barry K, Blanchette RA, Henrissat B, Martinez AT, Otillar R, Spatafora JW, Yadav JS, Aerts A, Benoit I, Boyd A, Carlson A, Copeland A, Coutinho PM, de Vries RP, Ferreira P, Findley K, Foster B, Gaskell J, Glotzer D, Gorecki P, Heitman J, Hesse C, Hori C, Igarashi K, Jurgens JA, Kallen N, Kersten P, Kohler A, Kues U, Kumar TK, Kuo A, LaButti K, Larrondo LF, Lindquist E, Ling A, Lombard V, Lucas S, Lundell T, Martin R, McLaughlin DJ, Morgenstern I, Morin E, Murat C, Nagy LG, Nolan M, Ohm RA, Patyshakuliyeva A, Rokas A, Ruiz-Duenas FJ, Sabat G, Salamov A, Samejima M, Schmutz J, Slot JC, St John F, Stenlid J, Sun H, Sun S, Syed K, Tsang A, Wiebenga A, Young D, Pisabarro A, Eastwood DC, Martin F, Cullen D, Grigoriev IV, Hibbett DS | 2012 | Wolfiporia cocos genome sequence | http://genome.jgi-psf.org/Wolco1/Wolco1.home.html | Publicly available at genome.jgi.doe.gov |
| Cubeta MA, Thomas E, Dean RA, Jabaji S, Neate SM, Tavantzis S, Toda T, Vilgalys R, Bharathan N, Fedorova-Abrams N, Pakala SB, Pakala SM, Zafar N, Joardar V, Losada L, Nierman WC. | 2014 | Rhizoctonia solani genome sequence | http://www.ebi.ac.uk/ena/data/view/GCA_000524645.1 | Publicly available at the EBI European Nucleotide Archive (accession no:GCA_000524645.1) |
| Eastwood DC, Floudas D, Binder M, Majcherczyk A, Schneider P, Aerts A, Asiegbu FO, Baker SE, Barry K, Bendiksby M, Blumentritt M, Coutinho PM, Cullen D, | 2011 | Serpula lacrymans genome sequence | http://genome.jgi-psf.org/SerlaS7_9_2/SerlaS7_9_2.home.html | Publicly available at genome.jgi-psf.org |

| | | | | |
|---|---|---|---|---|
| de Vries RP, Gathman A, Goodell B, Henrissat B, Ihrmark K, Kauserud H, Kohler A, LaButti K, Lapidus A, Lavin JL, Lee YH, Lindquist E, Lilly W, Lucas S, Morin E, Murat C, Oguiza JA, Park J, Pisabarro AG, Riley R, Rosling A, Salamov A, Schmidt O, Schmutz J, Skrede I, Stenlid J, Wiebenga A, Xie X, Kues U, Hibbett DS, Hoffmeister D, Hogberg N, Martin F, Grigoriev IV, Watkinson SC | | | | |
| Meinhardt LW, Costa GG, Thomazella DP, Teixeira PJ, Carazzolle MF, Schuster SC, Carlson JE, Guiltinan MJ, Mieczkowski P, Farmer A, Ramaraj T, Crozier J, Davis RE, Shao J, Melnick RL, Pereira GA, Bailey BA | 2014 | Moniliophthora roreri genome sequence | ftp://ftp.ncbi.nlm.nih.gov/genomes/all/GCA/000/488/995/GCA_000488995.1_M_roreri_MCA_2997_v1/ | Publicly available at www.ncbi.nlm.nih.gov.elis.tmu.edu.tw |
| Morin E, Kohler A, Baker AR, Foulongne-Oriol M, Lombard V, Nagy LG, Ohm RA, Patyshakuliyeva A, Brun A, Aerts AL, Bailey AM, Billette C, Coutinho PM, Deakin G, Doddapaneni H, Floudas D, Grimwood J, Hilden K, Kues U, Labutti KM, Lapidus A, Lindquist EA, Lucas SM, Murat C, Riley RW, Salamov AA, Schmutz J, Subramanian V, Wosten HA, Xu J, Eastwood DC, Foster GD, Sonnenberg AS, Cullen D, de Vries RP, Lundell T, Hibbett DS, Henrissat B, Burton KS, Kerrigan RW, Challen MP, Grigoriev IV, Martin F | 2012 | Agaricus bisporus genome sequence | http://genome.jgi.doe.gov/Agabi_varbisH97_2/Agabi_varbisH97_2.home.html | Publicly available at genome.jgi.doe.gov |
| Cisse OH, Almeida JM, Fonseca A, Kumar AA, Salojarvi J, Overmeyer K, Hauser PM, Pagni M | 2013 | Taphrina deformans genome sequence | http://genome.jgi.doe.gov/Tapde1_1/Tapde1_1.home.html | Publicly available at genome.jgi.doe.gov |
| Martin F, Aerts A, Ahrén D, Brun A, Danchin EG, Duchaussoy F, Gibon J, | 2008 | Tuber melanosporum genome sequence | http://genome.jgi.doe.gov/Tubme1/Tubme1.home.html | Publicly available at genome.jgi.doe.gov |

| | | | | |
|---|---|---|---|---|
| Kohler A, Lindquist E, Pereda V, Salamov A, Shapiro HJ, Wuyts J, Blaudez D, BuÃ{copyright, serif}e M, Brokstein P, CanbÃ¤ck B, Cohen D, Courty PE, Coutinho PM, Delaruelle C, Detter JC, Deveau A, DiFazio S, Duplessis S, Fraissinet-Tachet L, Lucic E, Frey-Klett P, Fourrey C, Feussner I, Gay G, Grimwood J, Hoegger PJ, Jain P, Kilaru S, LabbÃ {copyright, serif} J, Lin YC, LeguÃ {copyright, serif} V, Le Tacon F, Marmeisse R, Melayah D, Montanini B, Muratet M, Nehls U, Niculita-Hirzel H, Oudot-Le Secq MP, Peter M, Quesneville H, Rajashekar B, Reich M, Rouhier N, Schmutz J, Yin T, Chalot M, Henrissat B, KÃ1/4es U, Lucas S, Van de Peer Y, Podila GK, Polle A, Pukkila PJ, Richardson PM, RouzÃ{copyright, serif} P, Sanders IR, Stajich JE, Tunlid A, Tuskan G, Grigoriev IV | | | | |
| Marcet-Houben M, Ballester AR, de la Fuente B, Harries E, Marcos JF, Gonzalez-Candelas L, Gabaldon T | 2012 | Penicillium digitatum genome sequence | http://genome.jgi.doe. gov/Pendi1/Pendi1. home.html | Publicly available at genome.jgi.doe. gov |
| Nierman WC, Pain A, Anderson MJ, Wortman JR, Kim HS, Arroyo J, Berriman M, Abe K, Archer DB, Bermejo C, Bennett J, Bowyer P, Chen D, Collins M, Coulsen R, Davies R, Dyer PS, Farman M, Fedorova N, Feldblyum TV, Fischer R, Fosker N, Fraser A, Garcia JL, Garcia MJ, Goble A, Goldman GH, Gomi K, Griffith-Jones S, Gwilliam R, Haas B, Haas H, Harris D, Horiuchi H, Huang J, Humphray S, Jimenez J, Keller N, Khouri H, | 2005 | Aspergillus fumigatus genome sequence | http://genome.jgi.doe. gov/Aspfu1/Aspfu1. home.html | Publicly available at genome.jgi.doe. gov |

Kitamoto K, Kobayashi T, Konzack S, Kulkarni R, Kumagai T, Lafon A, Latge JP, Li W, Lord A, Lu C, Majoros WH, May GS, Miller BL, Mohamoud Y, Molina M, Monod M, Mouyna I, Mulligan S, Murphy L, O'Neil S, Paulsen I, Penalva MA, Pertea M, Price C, Pritchard BL, Quail MA, Rabbinowitsch E, Rawlins N, Rajandream MA, Reichard U, Renauld H, Robson GD, Rodriguez de Cordoba S, Rodriguez-Pena JM, Ronning CM, Rutter S, Salzberg SL, Sanchez M, Sanchez-Ferrero JC, Saunders D, Seeger K, Squares R, Squares S, Takeuchi M, Tekaia F, Turner G, Vazquez de Aldana CR, Weidman J, White O, Woodward J, Yu JH, Fraser C, Galagan JE, Asai K, Machida M, Hall N, Barrell B, Denning DW

| | | | | |
|---|---|---|---|---|
| Hane JK, Lowe RG, Solomon PS, Tan KC, Schoch CL, Spatafora JW, Crous PW, Kodira C, Birren BW, Galagan JE, Torriani SF, McDonald BA, Oliver RP | 2007 | Stagonospora nodorum genome sequence | http://genome.jgi.doe.gov/Stano2/Stano2.home.html | Publicly available at genome.jgi.doe.gov |
| Ohm RA, Feau N, Henrissat B, Schoch CL, Horwitz BA, Barry KW, Condon BJ, Copeland AC, Dhillon B, Glaser F, Hesse CN, Kosti I, LaButti K, Lindquist EA, Lucas S, Salamov AA, Bradshaw RE, Ciuffetti L, Hamelin RC, Kema GH, Lawrence C, Scott JA, Spatafora JW, Turgeon BG, de Wit PJ, Zhong S, Goodwin SB, Grigoriev IV | 2012 | Alternaria brassicicola genome sequence | http://genome.jgi.doe.gov/Altbr1/Altbr1.home.html | Publicly available at genome.jgi.doe.gov |
| Manning VA, Pandelova I, Dhillon B, Wilhelm LJ, Goodwin SB, Berlin AM, Figueroa M, Freitag | 2013 | Pyrenophora tritici-repentis genome sequence | https://www.broadinstitute.org/scientific-community/science/projects/fungal-genome-initiative/pyrenophora-genome- | Publicly available at www.broadinstitute.org |

| | | | | |
|---|---|---|---|---|
| M, Hane JK, Henrissat B, Holman WH, Kodira CD, Martin J, Oliver RP, Robbertse B, Schackwitz W, Schwartz DC, Spatafora JW, Turgeon BG, Yandava C, Young S, Zhou S, Zeng Q, Grigoriev IV, Ma LJ, Ciuffetti LM | | | | project |
| de Wit PJ, van der Burgt A, Okmen B, Stergiopoulos I, Abd-Elsalam KA, Aerts AL, Bahkali AH, Beenen HG, Chettri P, Cox MP, Datema E, de Vries RP, Dhillon B, Ganley AR, Griffiths SA, Guo Y, Hamelin RC, Henrissat B, Kabir MS, Jashni MK, Kema G, Klaubauf S, Lapidus A, Levasseur A, Lindquist E, Mehrabi R, Ohm RA, Owen TJ, Salamov A, Schwelm A, Schijlen E, Sun H, van den Burg HA, van Ham RC, Zhang S, Goodwin SB, Grigoriev IV, Collemare J, Bradshaw RE | 2012 | Dothistroma septosporum genome sequence | http://genome.jgi.doe. gov/Dotse1/Dotse1. home.html | Publicly available at genome.jgi.doe. gov |
| Ohm RA, Feau N, Henrissat B, Schoch CL, Horwitz BA, Barry KW, Condon BJ, Copeland AC, Dhillon B, Glaser F, Hesse CN, Kosti I, LaButti K, Lindquist EA, Lucas S, Salamov AA, Bradshaw RE, Ciuffetti L, Hamelin RC, Kema GH, Lawrence C, Scott JA, Spatafora JW, Turgeon BG, de Wit PJ, Zhong S, Goodwin SB, Grigoriev IV | 2012 | Pseudocercospora fijiensis genome sequence | ftp://ftp.ncbi.nlm.nih. gov/genomes/all/GCA/ 000/340/215/GCA_ 000340215.1_Mycfi2/ | Publicly available at ftp.ncbi.nlm.nih. gov |
| Goodwin SB, M'barek SB, Dhillon B, Wittenberg AH, Crane CF, Hane JK, Foster AJ, Van der Lee TA, Grimwood J, Aerts A, Antoniw J, Bailey A, Bluhm B, Bowler J, Bristow J, van der Burgt A, Canto-Canche B, Churchill AC, Conde-Ferraez L, Cools HJ, Cou- | 2011 | Zymoseptoria tritici genome sequence | http://genome.jgi.doe. gov/Mycgr3/Mycgr3. home.html | Publicly available at genome.jgi.doe. gov |

| | | | | |
|---|---|---|---|---|
| tinho PM, Csukai M, Dehal P, De Wit P, Donzelli B, van de Geest HC, van Ham RC, Hammond-Kosack KE, Henrissat B, Kilian A, Kobayashi AK, Koopmann E, Kourmpetis Y, Kuzniar A, Lindquist E, Lombard V, Maliepaard C, Martins N, Mehrabi R, Nap JP, Ponomarenko A, Rudd JJ, Salamov A, Schmutz J, Schouten HJ, Shapiro H, Stergiopoulos I, Torriani SF, Tu H, de Vries RP, Waalwijk C, Ware SB, Wiebenga A, Zwiers LH, Oliver RP, Grigoriev IV, Kema GH | | | | |
| de Wit PJ, van der Burgt A, Okmen B, Stergiopoulos I, Abd-Elsalam KA, Aerts AL, Bahkali AH, Beenen HG, Chettri P, Cox MP, Datema E, de Vries RP, Dhillon B, Ganley AR, Griffiths SA, Guo Y, Hamelin RC, Henrissat B, Kabir MS, Jashni MK, Kema G, Klaubauf S, Lapidus A, Levasseur A, Lindquist E, Mehrabi R, Ohm RA, Owen TJ, Salamov A, Schwelm A, Schijlen E, Sun H, van den Burg HA, van Ham RC, Zhang S, Goodwin SB, Grigoriev IV, Collemare J, Bradshaw RE | 2012 | Passalora fulva genome sequence | http://genome.jgi.doe.gov/Clafu1/Clafu1.home.html | Publicly available at genome.jgi.doe.gov |
| Jones L, Riaz S, Morales-Cruz A, Amrine KC, McGuire B, Gubler WD, Walker MA, Cantu D | 2014 | Erysiphe necator genome sequence | http://www.ncbi.nlm.nih.gov/Traces/wgs/?val=JNVN01#contigs | Publicly available at www.ncbi.nlm.nih.gov |
| van Kan JA, Stassen JH, Mosbach A, van der Lee TA, Faino L, Farmer AD, Papasotiriou D, Zhou S, Seidl MF, Cottam E, Edel D, Hahn M, Schwartz DC, Dietrich RA, Widdison S, Scalliet G | 2016 | Botrytis cinerea genome sequence | http://fungi.ensembl.org/Botrytis_cinerea/Info/Index | Publicly available at fungi.ensembl.org |
| Kohler A, Kuo A, Nagy LG, Morin E, | 2015 | Oidiodendron maius genome sequence | http://genome.jgi.doe.gov/Oidma1/Oidma1. | Publicly available at genome.jgi.doe. |

| | | | home.html | gov |
|---|---|---|---|---|
| Barry KW, Buscot F, Canback B, Choi C, Cichocki N, Clum A, Colpaert J, Copeland A, Costa MD, Dore J, Floudas D, Gay G, Girlanda M, Henrissat B, Herrmann S, Hess J, Hogberg N, Johansson T, Khouja HR, LaButti K, Lahrmann U, Levasseur A, Lindquist EA, Lipzen A, Marmeisse R, Martino E, Murat C, Ngan CY, Nehls U, Plett JM, Pringle A, Ohm RA, Perotto S, Peter M, Riley R, Rineau F, Ruytinx J, Salamov A, Shah F, Sun H, Tarkka M, Tritt A, Veneault-Fourrey C, Zuccaro A, Tunlid A, Grigoriev IV, Hibbett DS, Martin F | | | | |
| Geomyces destructans Sequencing Project, Broad Institute of Harvard and MIT | 2016 | Pseudogymnoascus destructans genome sequence | https://www.broadinstitute.org/fungal-genome-initiative/geomyces-destructans-genome-project | Publicly available at www.broadinstitute.org |
| Dean RA, Talbot NJ, Ebbole DJ, Farman ML, Mitchell TK, Orbach MJ, Thon M, Kulkarni R, Xu JR, Pan H, Read ND, Lee YH, Carbone I, Brown D, Oh YY, Donofrio N, Jeong JS, Soanes DM, Djonovic S, Kolomiets E, Rehmeyer C, Li W, Harding M, Kim S, Lebrun MH, Bohnert H, Coughlan S, Butler J, Calvo S, Ma LJ, Nicol R, Purcell S, Nusbaum C, Galagan JE, Birren BW | 2005 | Magnaporthe oryzae genome sequence | http://genome.jgi.doe.gov/Maggr1/Maggr1.home.html | Publicly available at genome.jgi.doe.gov |
| Berka RM, Grigoriev IV, Otillar R, Salamov A, Grimwood J, Reid I, Ishmael N, John T, Darmond C, Moisan MC, Henrissat B, Coutinho PM, Lombard V, Natvig DO, Lindquist E, Schmutz J, Lucas S, Harris P, Powlowski J, Bellemare A, Taylor D, Butler G, de Vries RP, Allijn IE, van den Brink J, Ushinsky S, Storms | 2011 | Myceliophthora thermophila genome sequence | http://genome.jgi.doe.gov/Spoth2/Spoth2.home.html | Publicly available at genome.jgi.doe.gov |

| | | | | | |
|---|---|---|---|---|---|
| R, Powell AJ, Paulsen IT, Elbourne LD, Baker SE, Magnuson J, Laboissiere S, Clutterbuck AJ, Martinez D, Wogulis M, de Leon AL, Rey MW, Tsang A | | | | | |
| Berka RM, Grigoriev IV, Otillar R, Salamov A, Grimwood J, Reid I, Ishmael N, John T, Darmond C, Moisan MC, Henrissat B, Coutinho PM, Lombard V, Natvig DO, Lindquist E, Schmutz J, Lucas S, Harris P, Powlowski J, Bellemare A, Taylor D, Butler G, de Vries RP, Allijn IE, van den Brink J, Ushinsky S, Storms R, Powell AJ, Paulsen IT, Elbourne LD, Baker SE, Magnuson J, Laboissiere S, Clutterbuck AJ, Martinez D, Wogulis M, de Leon AL, Rey MW, Tsang A | 2011 | Chaetomium globosum genome sequence | http://genome.jgi.doe.gov/Chagl_1/Chagl_1.home.html | Publicly available at genome.jgi.doe.gov |
| Faino L, Seidl MF, Datema E, van den Berg GC, Janssen A, Wittenberg AH, Thomma BP | 2015 | Verticilium dahliae genome sequence | http://www.ncbi.nlm.nih.gov/genome/?term=GCA_000952015.1 | Publicly available at NCBI Genome (accession no. GCA_000952015.1) |
| O'Connell RJ, Thon MR, Hacquard S, Amyotte SG, Kleemann J, Torres MF, Damm U, Buiate EA, Epstein L, Alkan N, Altmuller J, Alvarado-Balderrama L, Bauser CA, Becker C, Birren BW, Chen Z, Choi J, Crouch JA, Duvick JP, Farman MA, Gan P, Heiman D, Henrissat B, Howard RJ, Kabbage M, Koch C, Kracher B, Kubo Y, Law AD, Lebrun MH, Lee YH, Miyara I, Moore N, Neumann U, Nordstrom K, Panaccione DG, Panstruga R, Place M, Proctor RH, Prusky D, Rech G, Reinhardt R, Rollins JA, Rounsley S, Schardl CL, Schwartz DC, Shenoy N, Shirasu K, | 2012 | Colletotrichum higginsianum genome sequence | http://genome.jgi.doe.gov/Colhi1/Colhi1.home.html | Publicly available at genome.jgi.doe.gov |

| Sikhakolli UR, Stuber K, Sukno SA, Sweigard JA, Takano Y, Takahara H, Trail F, van der Does HC, Voll LM, Will I, Young S, Zeng Q, Zhang J, Zhou S, Dickman MB, Schulze-Lefert P, Ver Loren van Themaat E, Ma LJ, Vaillancourt LJ | | | | |
| --- | --- | --- | --- | --- |
| Colletotrichum Sequencing Project, Broad Institute of Harvard and MIT | 2016 | Colletotrichum graminicola genome sequence | https://www.broadinstitute.org/scientific-community/science/projects/fungal-genome-initiative/colletotrichum-genome-project | Publicly available at www.broadinstitute.org |
| Xiao G, Ying SH, Zheng P, Wang ZL, Zhang S, Xie XQ, Shang Y, St Leger RJ, Zhao GP, Wang C, Feng MG | 2012 | Beauveria bassiana genome sequence | http://genome.jgi.doe.gov/Beaba1/Beaba1.home.html | Publicly available at genome.jgi.doe.gov |
| Ma LJ, van der Does HC, Borkovich KA, Coleman JJ, Daboussi MJ, Di Pietro A, Dufresne M, Freitag M, Grabherr M, Henrissat B, Houterman PM, Kang S, Shim WB, Woloshuk C, Xie X, Xu JR, Antoniw J, Baker SE, Bluhm BH, Breakspear A, Brown DW, Butchko RA, Chapman S, Coulson R, Coutinho PM, Danchin EG, Diener A, Gale LR, Gardiner DM, Goff S, Hammond-Kosack KE, Hilburn K, Hua-Van A, Jonkers W, Kazan K, Kodira CD, Koehrsen M, Kumar L, Lee YH, Li L, Manners JM, Miranda-Saavedra D, Mukherjee M, Park G, Park J, Park SY, Proctor RH, Regev A, Ruiz-Roldan MC, Sain D, Sakthikumar S, Sykes S, Schwartz DC, Turgeon BG, Wapinski I, Yoder O, Young S, Zeng Q, Zhou S, Galagan J, Cuomo CA, Kistler HC, Rep M | 2010 | Fusarium graminearum genome sequence | http://genome.jgi.doe.gov/Fusgr1/Fusgr1.home.html | Publicly available at genome.jgi.doe.gov |
| Gao Q, Jin K, Ying SH, Zhang Y, Xiao G, Shang Y, Duan Z, Hu X, Xie XQ, Zhou G, Peng G, Luo Z, Huang W, Wang B, | 2011 | Metarhizium acridum genome sequence | http://genome.jgi.doe.gov/pages/dynamicOrganismDownload.jsf?organism=Metac1 | Publicly available at genome.jgi.doe.gov |

| | | | | |
|---|---|---|---|---|
| Fang W, Wang S, Zhong Y, Ma LJ, St Leger RJ, Zhao GP, Pei Y, Feng MG, Xia Y, Wang C | | | | |
| Croll D, Zala M, McDonald BA | 2013 | Zymoseptoria tritici variant call file | http://fungi.ensembl.org/Zymoseptoria_tritici/Info/Index | Publicly available at fungi.ensembl.org |
| Farr DF, Rossman AY | 2016 | List of hosts for fungal plant pathogens | http://nt.ars-grin.gov/fungaldatabases/fungushost/FungusHost.cfm | Publicly available at nt.ars-grin.gov |
| Spanu PD, Abbott JC, Amselem J, Burgis TA, Soanes DM, Stüber K, Ver Loren van Themaat E, Brown JK, Butcher SA, Gurr SJ, Lebrun MH, Ridout CJ, Schulze-Lefert P, Talbot NJ, Ahmadinejad N, Ametz C, Barton GR, Benjdia M, Bidzinski P, Bindschedler LV, Both M, Brewer MT, Cadle-Davidson L, Cadle-Davidson MM, Collemare J, Cramer R, Frenkel O, Godfrey D, Harriman J, Hoede C, King BC, Klages S, Kleemann J, Knoll D, Koti PS, Kreplak J, López-Ruiz FJ, Lu X, Maekawa T, Mahanil S, Micali C, Milgroom MG, Montana G, Noir S, O'Connell RJ, Oberhaensli S, Parlange F, Pedersen C, Quesneville H, Reinhardt R, Rott M, Sacristán S, Schmidt SM, Schön M, Skamnioti P, Sommer H, Stephens A, Takahara H, Thordal-Christensen H, Vigouroux M, Wessling R, Wicker T, Panstruga R. | 2010 | Blumeria graminis genome sequence | http://genome.jgi.doe.gov/Blugr1/Blugr1.home.html | Publicly available at genome.jgi.doe.gov |
| Duplessis S, Cuomo CA, Lin YC, Aerts A, Tisserant E, Veneault-Fourrey C, Joly DL, Hacquard S, Amselem J, Cantarel BL, Chiu R, Coutinho PM, Feau N, Field M, Frey P, Gelhaye E, Goldberg J, Grabherr MG, Kodira CD, Kohler A, Kües U, Lindquist EA, Lucas SM, | 2011 | Melampsora larici-populina genome sequence | http://genome.jgi.doe.gov/Mellp2_3/Mellp2_3.home.html | Publicly available at genome.jgi.doe.gov |

| | | | | |
|---|---|---|---|---|
| Mago R, Mauceli E, Morin E, Murat C, Pangilinan JL, Park R, Pearson M, Quesneville H, Rouhier N, Sakthikumar S, Salamov AA, Schmutz J, Selles B, Shapiro H, Tanguay P, Tuskan GA, Henrissat B, Van de Peer Y, Rouzé P, Ellis JG, Dodds PN, Schein JE, Zhong S, Hamelin RC, Grigoriev IV, Szabo LJ, Martin F. | | | | |
| Martin F, Aerts A, Ahrén D, Brun A, Danchin EG, Duchaussoy F, Gibon J, Kohler A, Lindquist E, Pereda V, Salamov A, Shapiro HJ, Wuyts J, Blaudez D, Buée M, Brokstein P, Canbäck B, Cohen D, Courty PE, Coutinho PM, Delaruelle C, Detter JC, Deveau A, DiFazio S, Duplessis S, Fraissinet-Tachet L, Lucic E, Frey-Klett P, Fourrey C, Feussner I, Gay G, Grimwood J, Hoegger PJ, Jain P, Kilaru S, Labbé J, Lin YC, Legué V, Le Tacon F, Marmeisse R, Melayah D, Montanini B, Muratet M, Nehls U, Niculita-Hirzel H, Oudot-Le Secq MP, Peter M, Quesneville H, Rajashekar B, Reich M, Rouhier N, Schmutz J, Yin T, Chalot M, Henrissat B, Kües U, Lucas S, Van de Peer Y, Podila GK, Polle A, Pukkila PJ, Richardson PM, Rouzé P, Sanders IR, Stajich JE, Tunlid A, Tuskan G, Grigoriev IV. | 2008 | Laccaria bicolor genome sequence | http://genome.jgi.doe.gov/Lacbi2/Lacbi2.home.html | Publicly available at genome.jgi.doe.gov |
| Amselem J, Cuomo CA, van Kan JA, Viaud M, Benito EP, Couloux A, Coutinho PM, de Vries RP, Dyer PS, Fillinger S, Fournier E, Gout L, Hahn M, Kohn L, Lapalu N, Plummer KM, Pradier JM, Quévillon E, Sharon A, Simon A, ten | 2011 | Gene expression data for Botrytis cinerea | http://urgi.versailles.inra.fr/Data/Transcriptome | Publicly available at urgi.versailles.inra.fr |

Have A, Tudzynski B, Tudzynski P, Wincker P, Andrew M, Anthouard V, Beever RE, Beffa R, Benoit I, Bouzid O, Brault B, Chen Z, Choquer M, Collémare J, Cotton P, Danchin EG, Da Silva C, Gautier A, Giraud C, Giraud T, Gonzalez C, Grossetete S, Güldener U, Henrissat B, Howlett BJ, Kodira C, Kretschmer M, Lappartient A, Leroch M, Levis C, Mauceli E, Neuvéglise C, Oeser B, Pearson M, Poulain J, Poussereau N, Quesneville H, Rascle C, Schumacher J, Ségurens B, Sexton A, Silva E, Sirven C, Soanes DM, Talbot NJ, Templeton M, Yandava C, Yarden O, Zeng Q, Rollins JA, Lebrun MH, Dickman M

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
