## [Decision Letter]

[Editors’ note: this article was originally rejected after discussions between the reviewers, but the authors were invited to resubmit after an appeal against the decision.]

Thank you for submitting your work entitled "Codon optimization underpins generalist parasitism in fungi" for consideration by *eLife*. Your article has been favorably evaluated by Detlef Weigel (Senior Editor) and three reviewers, one of whom, Claus Wilke, is a member of our Board of Reviewing Editors.

Our decision has been reached after consultation between the reviewers. Based on these discussions and the individual reviews below, we regret to inform you that your work will not be considered further for publication in *eLife*. There were three key issues that prompted us to reach this conclusion:

1) The manuscript is lacking a convincing "why". No convincing theory is presented for why codon bias should differ between generalists and specialists.

2) The main finding of the paper is a correlation between codon bias and host breadth. However, this correlation is confounded by the phylogenetic relationship between species, and whether the correlation would remain after this relationship is controlled for is unclear.

3) The manuscript is not well conceived and structured. A lot of potentially interesting material is presented in supporting information, while some of the information in the main manuscript seems tangential to the overall story. For example, the material in Figure 2 only establishes that codon bias is caused by selection pressure (an observation that is already widely accepted) but does not actually support the main story of the paper (that generalists experience different selection pressures than specialists).

*Reviewer #1:*

1) I am missing a clear, overarching hypothesis. Why do the authors think that codon bias should differ between specialist and generalist species? What is driving this effect?

2) I am not sure that the work of Figure 2 speaks to the overall hypothesis of the paper. Figure 2 demonstrates that two genomes with different codon biases likely experience different synonymous selection pressures. I think it is widely accepted these days that codon bias is often caused by selection, so we haven't learned much new from this analysis. Importantly, this work does not explain *why* codon bias patterns might be different in a generalist vs. a specialist.

Technical comments:

3) Results and Discussion, first paragraph: "A total of 22 species showed signatures…" Was this corrected for multiple testing?

4) Results and Discussion, second paragraph, Figure 1: Spearman correlation assumes that the points are independent. However, they are confounded by phylogeny. The authors should calculate the correlation on phylogenetic independent contrasts. Looking at the figure, I would expect the effect to be much weaker under independent contrasts.

5) Subsection “Codon optimization across the kingdom Fungi”: I don't understand this: "taking the average p value from 100 tests of sample size=500 and n=1000." More explanation is needed. Normally, one would do only one test. How is an average p value from multiple tests to be interpreted?

6) What is the point of Figure 2? I don't understand what this is meant to show.

7) It is unclear what is shown in Figure 3. I don't think the quantity shown on the y axis is defined.

*Reviewer #2:*

The manuscript by Badet et al. aims to describe patterns of codon adaptation in parasitic fungi and how they vary as a function of their host ranges. In addition to performing comparative genomic analyses across 45 species, they generate transcriptomic datasets to measure changes in mRNA and tRNA abundances depending on the host for a subset of species as well as perform simulation studies to estimate elongation times of codons across these species.

Overall it is quite a compelling piece of work and I highly recommend its publication. My primary concern is that the bulk of analyses that the authors have performed is tucked away in the supplement and barely described in the main text. In fact, some supplementary figures are not even cited. I feel the authors are shortchanging themselves by submitting this manuscript as a Short report. I would recommend they elaborate their explanations of the analyses already presented in the supplement and discuss them in more detail.

In terms of specific concerns – I was particularly struck by their Figure 2—figure – supplement 1B. A large number of synonymous mutations are derived from mutations in the first codon position. Only Leu and Arg are capable of such mutations. Does this mean that over half the synonymous mutations were occurring at these two amino acids?

*Reviewer #3:*

In this study, the authors examine the relationship between gene codon optimization and host range variations in parasitic fungi. The study raised the hypothesis that the degree of gene codon optimization in parasitic fungi is related to the host range variation. Parasitic fungi with high degrees of gene codon optimization are usually generalists while genes in specialist fungi are usually less codon-optimized.

Although this is a very interesting hypothesis, the conclusions, however, are not convincing due to a very problematic methodology used and are not consistent with the known mechanisms of host specificity determination of parasitic fungi.

The conclusions of this paper are heavily dependent on a previously developed method (dos Reis et al., 2004) to determine the degree of codon tRNA coadapation. Such a value called 'S' was assumed to reflect the translation selection of genes by codon usage biases. Such a model, however, is very problematic and its analysis results are not consistent with many known experimental results. For example, in the original 2004 study, among the 126 genomes analyzed by this model, the S values of only 36 genomes were found to be statistically different from zero, suggesting that for most genomes, there is no sign of translational selection by codon usage acting on their genomes. Obviously, this is very different from known experimental. For example, while S values of human and *Drosophila* indicating lack of selection, there are now very strong experimental evidence demonstrating a role for translation selection by codon usage biases. Same for Bacillus. As acknowledged by the authors of the 2004 paper, their model has severe limitations and cannot explain much of the variation observed. In the case of human and mouse genome, which should exhibit a very similar degree of translation selection by codon biases, has quite a big difference in S values. In the context of this paper, three of the non-parasitic fungi analyzed by the 2004 study, *S. cerevisiae, S. pombe*, and *N. crassa*, all have very high S values, which are not consistent with the conclusion of this paper. All these indicate that the S value model used is very problematic and do not reflect much of the known experimental results.

A brief reading of the original 2004 method calculating the optimized s-values show that although adenosine deamination was taken into account when calculating base pairing between tRNA and codons, the I:U base pairing was set to be more preferred than I:C based pairing. Obviously, this is not consistent with current knowledge that inosine preferentially base pairs with cytosine. On the other hand, the simple assumption that tRNA copy number will truthfully reflect tRNA expression will certainly introduce additional variations of the model (human vs. mice for example). Current tRNA sequencing methods do not help here due to cloning and sequencing biases. Obviously, there can be more issues with the model.

The host specificity determination of parasitic fungi is mostly known to be determined by expression of different effectors, which frequently are not conserved. Although the codon optimization of genes related to infection was examined in two species, effector genes was not studied. Even if codon optimization may affect the expression of effectors, it should only affect the degree of infection but not the host range. Based on my knowledge, the expression levels of many known effectors are usually low.

Because of these two major deficiencies, I feel that the current study is premature for publication.

[Editors’ note: what now follows is the decision letter after the authors submitted for further consideration.]

Thank you for resubmitting your work entitled "Codon optimization underpins generalist parasitism in fungi" for further consideration at *eLife*. Your revised article has been favorably evaluated by Detlef Weigel (Senior editor), a Reviewing editor, and two reviewers.

The manuscript has been improved but there are some remaining issues that need to be addressed before acceptance, as outlined below:

1) Our major remaining concern with the manuscript lies in the "why" question brought up during the previous review. The arguments made are weak and hand-wavy. You claim that since generalists have longer genes than specialists and as a result might take longer to translate, there is stronger selection to improve their codon usage. This begs the question, what evolutionary forces are influencing gene lengths? Are gene-lengths in specialists under stronger selection to reduce in size or in generalists to increase their sizes and why?

Further, the results in Figure 5 suggest that codon optimization of secreted proteins may explain the host range of fungal parasites. However, the host range can simply be determined by robustness of overall cell metabolism/growth. To strengthen your conclusions, we suggest that you perform pathway-specific analysis and compare to that of the secreted proteins.

2) We appreciate that you have pursued several different approaches to correct for phylogeny, and we consider your results now reliable. However, throughout the manuscript there remain several places where you provide standard correlations, with uncorrected p values. All these should be replaced with correlations over phylogenetic contrasts. Specifically, these are in subsection “Codon optimization correlates with fungal parasites host range”, first paragraph and second paragraphs.

3) In several places throughout the manuscript, you list p values but don't state the test that was performed. Please state the test every time. Also, you are inconsistent in when you do and don't run a test. For example, in the first paragraph of the subsection “Long proteins encoded by the genome of generalist fungi likely increase natural selection on codon optimization” you list a p value but you don't do so in any of the other length comparisons in the following lines.

---

## [Author Response]

[Editors’ note: the author responses to the first round of peer review follow.]

*Our decision has been reached after consultation between the reviewers. Based on these discussions and the individual reviews below, we regret to inform you that your work will not be considered further for publication in eLife. There were three key issues that prompted us to reach this conclusion:*

*1) The manuscript is lacking a convincing "why". No convincing theory is presented for why codon bias should differ between generalists and specialists.*

The manuscript has been re-organized and amended to clearly state our “why” hypothesis: we propose that the longer proteins encoded by the genome of generalist fungi increase natural selection on the optimization of codons in these species. The rationale for this hypothesis is presented in the Introduction. Briefly, we observed that the genome of generalists typically encode longer proteins than that of specialists, probably in relation to requirements of their lifestyle. Overall, the time required for protein synthesis by the translation machinery increases with protein length. Therefore, near parasites maximal growth rate, the structural constraints on the translation machinery are higher in generalist than in specialist cells. This structural constraint can be alleviated by increasing protein synthesis rate via codon optimization. We also highlight the consistency of our hypothesis with the “jack of all trades, master of none” theoretical model for the evolution of generalism. Indeed, natural selection on the optimization of protein translation is expected to increase parasites fitness in average on multiple hosts rather than specifically on a single or a few hosts (Introduction, last paragraph).

As a first approach to test this hypothesis, we compared structural constraints on protein synthesis in specialist and generalist fungi and their impact on maximal cell growth rates. This analysis is based on an in silico model of the cellular translation machinery and proteome properties of sequenced fungal parasites. We conclude from this modeling approach that “longer proteins, especially secreted proteins, encoded in the genome of generalist fungi limits maximal cell growth rates compared to specialist fungi. [Our model] also shows that codon optimization can support the secretion of more complex proteins with limited growth penalty. […] We therefore expect natural selection on the optimization of codons to be stronger in generalist parasites than in host specialists.”

*2) The main finding of the paper is a correlation between codon bias and host breadth. However, this correlation is confounded by the phylogenetic relationship between species, and whether the correlation would remain after this relationship is controlled for is unclear.*

We have included explicit controls for phylogenetic signal in the revised manuscript. First, we used phylogeny and ancestral state reconstruction to show that codon optimization and host range co-evolved multiple times independently in the fungal Kingdom. Second, we show that the correlation between fungal parasites host range and codon optimization is detected both at the Kingdom and at the Phylum level. Our analysis of *S. sclerotiorum* and *Z. tritici* natural populations further shows that codon optimization can be detected at the infra-specific level in generalist but not in specialist species. Third, we used Blomberg’s K, Pagel’s λ and the phylosignal correlogram approach as quantitative measures of phylogenetic signal in codon optimization along the fungal phylogeny. Fourth, we verified the significance of the correlation between host range and genome scale codon optimization using Felsenstein’s phylogenetic independent contrasts (doi:10.1086/284325). These four methods allowed to unambiguously exclude an impact of phylogenetic relationships in the correlation we observed between codon optimization and host range.

*3) The manuscript is not well conceived and structured. A lot of potentially interesting material is presented in supporting information, while some of the information in the main manuscript seems tangential to the overall story. For example, the material in Figure 2 only establishes that codon bias is caused by selection pressure (an observation that is already widely accepted) but does not actually support the main story of the paper (that generalists experience different selection pressures than specialists).*

We have substantially revised the manuscript and improved its structure to clarify the underlying hypotheses and rationale for the analyses presented. We have reduced supporting information to the minimum required to support the conclusions of the paper (mostly source data). We notably clarify that the analyses presented in former Figure 2 aim at deciphering the molecular bases underlying codon optimization in generalist fungi.

We show that natural selection reduced the frequency of synonymous SNPs fixed in optimal codons in the genome of a generalist but not a specialist parasite. This first demonstrates that codon optimization associates with generalism at the infra-specific level, in addition to the Phylum and Kingdom levels. To our knowledge, comparative analyses describing evolutionary processes conserved from the infra-specific to the Kingdom level remain scarce. Second, this section of the manuscript connects codon optimization and broad host range through a molecular mechanism, indicating that the observed correlation is causal. Finally, by highlighting purifying selection acting on optimal codons in generalist but not in specialist parasites, this work provides further evidence that generalists experience different selection pressure than specialists. The significant changes we brought to this revised version of the manuscript should make these conclusions and implications clearer.

*Reviewer #1:*

*1) I am missing a clear, overarching hypothesis. Why do the authors think that codon bias should differ between specialist and generalist species? What is driving this effect?*

As mentioned in our response to the Editor, we have significantly edited the manuscript to make our hypothesis clear. We propose that selection on codon optimization would be stronger in generalists than in specialists because the genomes of generalist parasites encode longer proteins increasing the structural constraints on the translation machinery in these species. These structural constraints can be alleviated by codon optimization. This hypothesis derives from previous genomic studies of generalist and specialist fungi (see Introduction) and is supported by our analysis of an in silico model of the cell translation machinery calibrated with the properties of proteins from sequenced fungal parasites (see Results subsection “Long proteins encoded by the genome of generalist fungi likely increase natural selection on codon optimization”).

More specifically, our analysis shows that in average, secreted proteins are ~14.8% longer in generalist than in specialist fungi. Our modeling approach shows that differences in protein lengths imply higher codon decoding rates in generalists to achieve similar cellular growth rates as specialists. Therefore, codon optimization provides a mean for generalists to compete with specialist fungi and be maintained through evolution. This observation does not exclude other possible causes but is sufficient to expect natural selection on the optimization of codons to be stronger in generalist parasites than in host specialists.

*2) I am not sure that the work of Figure 2 speaks to the overall hypothesis of the paper. Figure 2 demonstrates that two genomes with different codon biases likely experience different synonymous selection pressures. I think it is widely accepted these days that codon bias is often caused by selection, so we haven't learned much new from this analysis. Importantly, this work does not explain why codon bias patterns might be different in a generalist vs. a specialist.*

The analyses reported in former Figure 2 (now Figure 4) aim at unraveling how natural selection drives codon optimization in generalists, rather than why this is the case. We explicitly state this objective in the first paragraph of the subsection “Biased SNP patterns underpin with codon optimization in the generalist parasite Sclerotinia sclerotiorum”. In Figure 4, we show first that codon optimization associates with generalism from the infra-specific to the Kingdom level. The evolutionary processes revealed in our work therefore stand out for their remarkable broad scope in the tree of life. Second, we connect codon optimization and broad host range through the molecular mechanism of biased synonymous SNPs on optimal codons. This supports a causal relationship between codon optimization and generalism. Finally, we detect purifying selection on optimal codon in the genome of generalist but not specialist fungi, reinforcing the finding that natural selection differs in these species. We conclude that “these findings identify biased synonymous substitutions as a link between generalism and codon optimization. This analysis is independent of codon usage indices and shows that selection for average performances on multiple hosts is reflected in global trends of genome evolution.” In the revised manuscript, we address the question why natural selection on codon optimization is expected to differ in generalist and specialists in the Introduction and results reported in Figure 1 (see response to reviewer 1 comment #1).

*Technical comments:*

*3) Results and Discussion, first paragraph: "A total of 22 species showed signatures…" Was this corrected for multiple testing?*

We have used Bonferroni correction to control for multiple testing in the revised version of the manuscript.

*4) Results and Discussion, second paragraph, Figure 1: Spearman correlation assumes that the points are independent. However, they are confounded by phylogeny. The authors should calculate the correlation on phylogenetic independent contrasts. Looking at the figure, I would expect the effect to be much weaker under independent contrasts.*

In the revised manuscript we have explicitly controlled for phylogenetic signal in the correlation between codon usage and host range. We provide four lines of evidence to demonstrate that this correlation is independent of phylogeny: (i) we used ancestral state reconstruction to show that codon optimization and host range co-evolved multiple times independently in the fungal Kingdom; (ii) we show that the correlation is detected at the Phylum level, (iii) we used three different quantitative measures of phylogenetic signal that all excluded any significant impact of phylogenetic relationships in the correlation between codon optimization and host range, and (iv) we verified the significance of the correlation between host range and codon optimization using Felsenstein’s phylogenetic independent contrasts (doi:10.1086/284325) and found correlation coefficients ~0.6 (p-val<2.8 10^-05^).

*5) Subsection “Codon optimization across the kingdom Fungi”: I don't understand this: "taking the average p value from 100 tests of sample size=500 and n=1000." More explanation is needed. Normally, one would do only one test. How is an average p value from multiple tests to be interpreted?*

We have used the p-value reported by a single test in the revised version of the manuscript.

*6) What is the point of Figure 2? I don't understand what this is meant to show.*

We have clarified the objectives of this approach as well as its implementation in our revised manuscript (subsection “Natural selection drives codon optimization in generalist fungal parasites”, first paragraph). We conclude from this approach that “SNP patterns determined experimentally converge towards increased codon optimality in *S. sclerotiorum* genome” and that “patterns of evolution towards increased codon optimality were detected in *S. sclerotiorum* but not in *Z. tritici* populations, and deviate significantly from neutral evolution”. This contributes to the unraveling of the molecular mechanisms underlying codon optimization in generalists and to the demonstration of a causal relationship between codon optimization and host range.

*7) It is unclear what is shown in Figure 3. I don't think the quantity shown on the y axis is defined.*

Former Figure 3 has been removed from the revised version of the manuscript.

*Reviewer #2:*

*The manuscript by Badet et al. aims to describe patterns of codon adaptation in parasitic fungi and how they vary as a function of their host ranges. In addition to performing comparative genomic analyses across 45 species, they generate transcriptomic datasets to measure changes in mRNA and tRNA abundances depending on the host for a subset of species as well as perform simulation studies to estimate elongation times of codons across these species.*

*Overall it is quite a compelling piece of work and I highly recommend its publication. My primary concern is that the bulk of analyses that the authors have performed is tucked away in the supplement and barely described in the main text. In fact, some supplementary figures are not even cited. I feel the authors are shortchanging themselves by submitting this manuscript as a Short report. I would recommend they elaborate their explanations of the analyses already presented in the supplement and discuss them in more detail.*

We are grateful for the reviewer’s assessment that our study represents a compelling piece of work the publication of which was recommended. We have taken the suggestion of reviewer #2 to remove the less useful supplementary material and to expand on the hypotheses, rationale and interpretation of our analyses by submitting our revised manuscript as an Article.

*In terms of specific concerns – I was particularly struck by their Figure 2—figure supplement 1B. A large number of synonymous mutations are derived from mutations in the first codon position. Only Leu and Arg are capable of such mutations. Does this mean that over half the synonymous mutations were occurring at these two amino acids?*

This unexpected value was due to a mistake in the way gene orientation was taken into account in our original script. We are sorry that this error escaped our scrutiny in the first version of the manuscript and thank the reviewer for pointing it out. The figure has been corrected accordingly. This information is not used in any other analysis of this work so that all other conclusions regarding SNP patterns remain valid.

*Reviewer #3:*

In this study, the authors examine the relationship between gene codon optimization and host range variations in parasitic fungi. The study raised the hypothesis that the degree of gene codon optimization in parasitic fungi is related to the host range variation. Parasitic fungi with high degrees of gene codon optimization are usually generalists while genes in specialist fungi are usually less codon-optimized.

*Although this is a very interesting hypothesis, the conclusions, however, are not convincing due to a very problematic methodology used and are not consistent with the known mechanisms of host specificity determination of parasitic fungi.*

*The conclusions of this paper are heavily dependent on a previously developed method (dos Reis et al., 2004) to determine the degree of codon tRNA coadapation. Such a value called 'S' was assumed to reflect the translation selection of genes by codon usage biases. Such a model, however, is very problematic and its analysis results are not consistent with many known experimental results. For example, in the original 2004 study, among the 126 genomes analyzed by this model, the S values of only 36 genomes were found to be statistically different from zero, suggesting that for most genomes, there is no sign of translational selection by codon usage acting on their genomes. Obviously, this is very different from known experimental. For example, while S values of human and Drosophila indicating lack of selection, there are now very strong experimental evidence demonstrating a role for translation selection by codon usage biases. Same for Bacillus. As acknowledged by the authors of the 2004 paper, their model has severe limitations and cannot explain much of the variation observed. In the case of human and mouse genome, which should exhibit a very similar degree of translation selection by codon biases, has quite a big difference in S values.*

The calculation of S value involves all predicted genes in a genome. A positive S signal therefore suggests that selection for codon optimization is high and targets a significant proportion of genes in a given genome. As mentioned by Dos Reis et al. (2004), highly expressed genes with biased codon patters represent only 4.6% of *Bacillus subtilis* genome, probably explaining why the S signal at the whole genome scale was weak in this species. The authors conclude that “small S-value for a whole genome means that translational selection might be negligible at a genomic scale, but it can nonetheless have a strong effect on smaller scales, such as particular gene sets”. A subset of genes is indeed likely under translational selection in host specialists, although this subset is too small to increase S significantly. We have carefully edited the manuscript to clarify that conclusions based on S value apply at the whole genome scale (e.g. subsection “Codon optimization correlates with fungal parasites host range”, first and second paragraphs), and we have added the following word of caution: “Translational selection may nevertheless be active in the genome of specialists but on limited gene sets only, resulting in a low signal in genome-scale analyses” (Discussion).

*In the context of this paper, three of the non-parasitic fungi analyzed by the 2004 study, S. cerevisiae, S. pombe, and N. crassa, all have very high S values, which are not consistent with the conclusion of this paper. All these indicate that the S value model used is very problematic and do not reflect much of the known experimental results.*

Our work does not imply that non parasitic fungi should have low S values. We have included a small set of non-parasitic fungi in our work to support an increase of S in generalist lineages, which required identifying closely related genomes with lower S values and the reconstruction of ancestral states (Figure 3). We have added the following sentence to clarify this point: “[Lower codon optimization in non parasitic fungi] does not exclude that some lineages of non-parasitic fungi could have evolved high codon optimization, but supports the view that codon optimization increased after the divergence of generalist parasites” (Results).

*A brief reading of the original 2004 method calculating the optimized s-values show that although adenosine deamination was taken into account when calculating base pairing between tRNA and codons, the I:U base pairing was set to be more preferred than I:C based pairing. Obviously, this is not consistent with current knowledge that inosine preferentially base pairs with cytosine. On the other hand, the simple assumption that tRNA copy number will truthfully reflect tRNA expression will certainly introduce additional variations of the model (human vs. mice for example). Current tRNA sequencing methods do not help here due to cloning and sequencing biases. Obviously, there can be more issues with the model.*

To our knowledge, measures of the adaptation of genes to the tRNA pool (such as S and the tRNA adaptation index) are among the most widely used methods these days to study codon usage bias. In a recent cross-species analysis, Sabi and Tuller (doi: 10.1093/dnares/dsu017) reported that species-specific s-values outperformed the original optimized s-values in non-fungal organisms, but not in fungi. Accordingly, they determined species-specific s-values for fungi very similar to dos Reis et al. 2004 optimized s-values. Besides, we have determined tRNA expression in *S. sclerotiorum* using Illumina sequencing approach to verify in this organism that tRNA copy number correlate with tRNA expression. For these reasons, we do not expect strong methodological bias in this approach.

Nevertheless, we agree that tRNA based methods have limitations for the analysis of codon optimization. To circumvent them, we used in our revised manuscript three other complimentary approaches: First, we inferred genome-wide codon optimization based on the Codon Adaptation Index (Sharp and Li, Nucleic acids research 1987) using ribosomal proteins from each genome as reference sets. This method does not rely on knowledge of the tRNA pool. Second, we inferred genome-wide codon optimization based on the recently developed self-consistent normalized Relative Codon Adaptation index (scnRCA, O’Neill et al. Plos One 2013). This method does not rely on a reference gene set. Finally, we compared codon optimization in core ortholog gene sets, to exclude possible biases due to the completeness of genome assembly. It should be noted that bias due to genome assembly could be excluded in our other analyses (see subsection “Codon optimization correlates with fungal parasites host range”, last paragraph). The four approaches converged toward the conclusion that codon optimization correlates with fungal parasites host range, supporting the robustness of this finding.

*The host specificity determination of parasitic fungi is mostly known to be determined by expression of different effectors, which frequently are not conserved. Although the codon optimization of genes related to infection was examined in two species, effector genes was not studied. Even if codon optimization may affect the expression of effectors, it should only affect the degree of infection but not the host range. Based on my knowledge, the expression levels of many known effectors are usually low.*

We agreed that “The host specificity determination of parasitic fungi is mostly known to be determined by expression of different effectors, which frequently are not conserved” in the context of gene-for-gene interactions, such as observed for host specialist parasites. It is however unclear whether effectors are the only determinants of host range and how crucial they are in the case of broad host range parasites. It is notable that the ability to derive nutrients from host tissues is critical for some broad host range parasites (e.g. Gesbert et al. 2014 DOI:10.1111/cmi.12227). For plant pathogens this involves the degradation of the plant cell wall by secreted enzymes. The expression level of such genes may be high during host colonization. Furthermore, the complete repertoire of effectors in not known for most fungal pathogens. For these reasons we chose not to restrict our analysis to effectors (or candidate effectors). Instead we used genes induced during host colonization and predicted secreted protein as genes universally associated with host colonization. Admittedly, codon optimization is unlikely to allow a parasite to evade recognition by a host resistance gene. It may however increase parasites fitness to enable the colonization of different hosts. We have clarified this point in the Discussion (last paragraph), concluding that “Codon optimization may affect the expression of effectors and other virulence factors, modifying the degree of infection or enabling the colonization of new hosts opposing quantitative disease resistance mechanisms.”

[Editors’ note: the author responses to the re-review follow.]

*The manuscript has been improved but there are some remaining issues that need to be addressed before acceptance, as outlined below:*

*1) Our major remaining concern with the manuscript lies in the "why" question brought up during the previous review. The arguments made are weak and hand-wavy. You claim that since generalists have longer genes than specialists and as a result might take longer to translate, there is stronger selection to improve their codon usage. This begs the question, what evolutionary forces are influencing gene lengths? Are gene-lengths in specialists under stronger selection to reduce in size or in generalists to increase their sizes and why?*

*Further, the results in Figure 5 suggest that codon optimization of secreted proteins may explain the host range of fungal parasites. However, the host range can simply be determined by robustness of overall cell metabolism/growth. To strengthen your conclusions, we suggest that you perform pathway-specific analysis and compare to that of the secreted proteins.*

The question of the evolution of gene length is an interesting field of research that reaches beyond the scope of our study. We have included elements of discussion on this point in our revised manuscript by stating that “Gene length increases in relation to evolutionary age, with conserved genes under purifying selection generally being longer and harboring higher frequency of optimal codons (Prat et al., 2009; Wolf et al., 2009). Neofunctionalization in duplicated genes has also been associated with increased gene length (He and Zhang, 2005). Duplications frequently underlie domain loss and gain in eukaryotic proteins, which is an important mechanism for the evolution of new functions (Buljan et al., 2010; Peisajovich et al., 2010).” In the context of our work, the possible implication is that “Natural selection may promote protein domains recombination to increase the versatility of fungal proteins functions and thereby contribute to host range expansion”.

Our analyses show that codon optimization of secreted proteins is stronger in the genome of generalist parasites compared to specialists. We show that it is also the case for host-induced genes, regardless of whether they encode secreted or non-secreted proteins (Figure 5). We agree that additional genomic properties likely contributed to the success of generalist fungi. We have taken the suggestion to test for codon optimization in multiple cellular processes such as metabolism. The result of this analysis is provided in Figure 5. We have added the following paragraph to describe these results: “Increased codon optimization in generalists was not only clear for genes encoding secreted proteins and host-induced genes, but also across their entire genome (‘other genes’ in Figure 5). […] Conversely, GOs related to transcription, transposable elements and phosphorelay signal transduction were better optimized in specialists than in generalists.”

The subsection “Codon optimization and genes function” describes the methods used for this analysis. The findings from this analysis may be related to the emergence of new protein functions through domain recombination and gene length expansion as discussed in the third paragraph of the Discussion. The importance of codon optimization in the evolution of these innovations, deduced from our work and current knowledge of molecular host-parasite interactions, is then discussed in the last paragraph of the Discussion.

We are grateful for these suggestions that prompted us to complete our analyses and add elements of context to the Discussion section of the manuscript. These revisions greatly clarified the expected functional implications of codon optimization in generalists.

*2) We appreciate that you have pursued several different approaches to correct for phylogeny, and we consider your results now reliable. However, throughout the manuscript there remain several places where you provide standard correlations, with uncorrected p values. All these should be replaced with correlations over phylogenetic contrasts. Specifically, these are in subsection “Codon optimization correlates with fungal parasites host range”, first paragraph and second paragraphs.*

We have replaced uncorrected correlations by correlations under phylogenetic independent contrasts in the first and second paragraphs of the subsection “Codon optimization correlates with fungal parasites host range”. To preserve the flow of the manuscript, we have moved the statement “We verified the significance of the correlation between host range and genome scale codon optimization using Felsenstein’s phylogenetic independent contrasts, and obtained Spearman rho of 0.6 (p-val=1.5 10-05)” to the first paragraph of the aforementioned subsection. We updated the correlation coefficients and p-values used in Figure 2 with correlations under phylogenetic independent contrasts.

*3) In several places throughout the manuscript, you list p values but don't state the test that was performed. Please state the test every time. Also, you are inconsistent in when you do and don't run a test. For example, in the first paragraph of the subsection “Long proteins encoded by the genome of generalist fungi likely increase natural selection on codon optimization” you list a p value but you don't do so in any of the other length comparisons in the following lines.*

We have added statement on the test used and p-values calculated wherever applicable.